# Annealing Self-Distillation Rectification Improves Adversarial Training

**Yu-Yu Wu, Hung-Jui Wang, Shang-Tse Chen**
National Taiwan University
{r10922018,r10922061,stchen}@csie.ntu.edu.tw

## Abstract

In standard adversarial training, models are optimized to fit invariant one-hot labels for adversarial data when the perturbations are within allowable budgets. However, the overconfident target harms generalization and causes the problem of robust overfitting. To address this issue and enhance adversarial robustness, we analyze the characteristics of robust models and identify that robust models tend to produce smoother and well-calibrated outputs. Based on the observation, we propose a simple yet effective method, **A**nnealing Self-**D**istillation **R**ectification (ADR), which generates soft labels as a better guidance mechanism that reflects the underlying distribution of data. By utilizing ADR, we can obtain rectified labels that improve model robustness without the need for pre-trained models or extensive extra computation. Moreover, our method facilitates seamless plug-and-play integration with other adversarial training techniques by replacing the hard labels in their objectives. We demonstrate the efficacy of ADR through extensive experiments and strong performances across datasets.

## 1 Introduction

Deep Neural Network (DNN) has been shown to exhibit susceptibility to adversarial attacks (Szegedy et al., 2014), wherein intentionally crafted imperceptible perturbations introduced into the original input cause the model's predictions to be altered. Among various defense methods (Kurakin et al., 2017; Liao et al., 2018; Wong & Kolter, 2018), Adversarial Training (AT) (Madry et al., 2018) stands out as one of the most effective techniques (Athalye et al., 2018; Uesato et al., 2018) to enhance DNN's adversarial robustness. However, while AT has proven effective in countering adversarial attacks, it is not immune to the problem of robust overfitting (Rice et al., 2020). AT constrains the model to generate consistent output when subjected to perturbations within an $\epsilon$ attack budget. However, manually assigned hard labels are noisy (Dong et al., 2022b;a) for AT since they fail to reflect shifts in the data distribution. Minimizing the adversarial training loss results in worse generalization ability on the test data. To address this issue, several approaches have been proposed, including label smoothing (Pang et al., 2021), consistency regularization (Dong et al., 2022b; Tarvainen & Valpola, 2017), and knowledge distillation (Chen et al., 2021; Cui et al., 2021; Goldblum et al., 2020a; Zhu et al., 2022; Zi et al., 2021; Zhao et al., 2022), which create a smoother objective function to alleviate the problem of robust overfitting. However, these approaches often assume uniform label noise (Pang et al., 2021), require consistency in the model's output over time (Dong et al., 2022b; Tarvainen & Valpola, 2017), or depend on an additional robust model trained in advance (Chen et al., 2021; Cui et al., 2021; Goldblum et al., 2020a). None of these methods directly focus on designing a well-rectified target without the need for pre-trained models.

To enhance AT, we investigate the characteristics that distinguish robust models from non-robust ones by analyzing the disparity in output distributions. Our findings indicate that robust models should possess good calibration ability, which is manifested by a lower average confidence level when it is likely to make errors. In addition, robust models' output distribution should remain consistent for the clean data and its adversarial counterpart. Based on this observation, we propose a novel approach called **A**nnealing Self-**D**istillation **R**ectification (ADR), which interpolates the outputs from model weight's momentum encoder with one-hot targets to generate noise-aware labels that reflect the underlying distribution of data. The intuition behind this method is that if an image

is similar to the other classes at once, we should assign a higher probability to those classes but still maintain the dominant probability for the actual class to ensure the final prediction is not altered.

The weight momentum encoding scheme, also known as Mean Teacher, is a widely used technique in semi-supervised (Tarvainen & Valpola, 2017) and self-supervised (Grill et al., 2020; Caron et al., 2021) learning that involves maintaining exponential moving average (EMA) of weights on the trained model. The self-distillation EMA also serves as a Weight Average (WA) (Garipov et al., 2018) method, which smoothes the loss landscape to enhance robustness (Izmailov et al., 2018; Chen et al., 2021; Gowal et al., 2020). To ensure our model generates well-calibrated results that reflect inter-class relations within examples, we introduce softmax temperature (Hinton et al., 2015) to scale outputs from the EMA model. Initially, the temperature is set to a larger value producing uniform distribution but gradually decreases following a cosine annealing schedule. This dynamic adjustment of temperature is undertaken in recognition of the EMA model's capacity to represent inter-class relationships improves over the course of training. We introduce an interpolation strategy to ensure the true class label consistently maintains the highest probability in targets. Notably, the interpolation factor experiences progressive increments as a reflection of our growing confidence in the precision and robustness of the momentum encoder. In summary, our contributions are:

- We conduct a comprehensive analysis of the robust models' output properties. Our investigation confirms the calibration ability of robust models. Additionally, we observe that robust models maintain output consistency on benign data and its adversarial counterpart, which motivates us to design a unified rectified label to enhance the efficacy of adversarial defense. For samples within the $l_p$ norm neighborhood of a given input, they should be associated with a single smooth and rectified target in adversarial training.

- We propose **A**nnealing Self-**D**istillation **R**ectification (ADR), a simple yet effective technique that leverages noise-aware label reformulation to refine the original one-hot target. Through this method, we obtain well-calibrated results without requiring pre-trained models or extensive extra computational resources. Additionally, ADR can be incorporated into other adversarial training algorithms at ease by substituting the hard label in their objectives, thus enabling a seamless plug-and-play integration.

- Our experimental results across multiple datasets demonstrate the efficacy of ADR in improving adversarial robustness. Substitute the hard labels with well-calibrated ones generated by ADR alone can achieve remarkable gains in robustness. When combined with other AT tricks (WA, AWP), ADR further outperforms the state-of-the-art results on CIFAR-100 and TinyImageNet-200 datasets with various architectures.

## 2  RELATED WORK

Adversarial training (AT) has been demonstrated to be effective in enhancing the white box robustness of DNN (Croce et al., 2021). PGD-AT (Madry et al., 2018), which introduces worst-case inputs during training, has been the most popular approach for improving robustness. An alternative AT method, TRADES (Zhang et al., 2019), provides a systematic approach to regulating the trade-off between natural accuracy and robustness and has yielded competitive results across multiple datasets. Despite the efficacy, AT often suffers from robust overfitting (Rice et al., 2020). Below, we summarize works that address the issue of robust overfitting by reforming the label. We also provide an extra survey on works aiming to mitigate the issue with other methods in Appendix A.

**Rectify labels in AT.**  AT can smooth the predictive distributions by increasing the likelihood of the target around the $\epsilon$-neighborhood of the observed training examples (Lakshminarayanan et al., 2017). A recent study by Grabinski et al. (2022) has shown that robust models produced by AT tend to exhibit lower confidence levels than non-robust models, even when evaluated on clean data. Due to the substantial differences in output distributions between robust and standard models, using one-hot labels, which encourage high-confidence predictions on adversarial examples, may not be optimal. Dong et al. (2022b) and Dong et al. (2022a) have demonstrated that one-hot labels are noisy in AT, as they are inherited from clean examples while the data had been distorted by attacks. The mismatch between the assigned labels and the true distributions can exacerbate overfitting compared to standard training. Rectifying labels is shown to be effective in addressing the issue of robust overfitting in AT (Dong et al., 2022a). Label Smoothing (LS) (Szegedy et al., 2016) is a technique

that softens labels by combining one-hot targets and a uniform distribution. By appropriately choosing the mixing factor, mild LS can enhance model robustness while calibrating the confidence of the trained model (Pang et al., 2021; Stutz et al., 2020). However, overly smoothing labels in a data-blind manner can diminish the discriminative power of the model (Müller et al., 2019; Paleka & Sanyal, 2022) and make it susceptible to gradient masking (Athalye et al., 2018). Prior works (Chen et al., 2021; Cui et al., 2021; Goldblum et al., 2020a; Zhu et al., 2022; Zi et al., 2021; Zhao et al., 2022) have utilized Knowledge Distillation (KD) to generate data-driven soft labels, outperforming baseline approaches. Temporal Ensembling (TE) (Dong et al., 2022b) and Mean Teacher (MT) (Zhang et al., 2022) have applied consistency loss into training objectives, thus preventing overconfident predictions through consistency regularization. More recently, Dong et al. (2022a) have employed a pre-trained robust model to reform training labels, addressing label noise in AT.

## 3 PRELIMINARIES

### 3.1 ADVERSARIAL TRAINING (AT)

Given a dataset $\mathcal{D} = \{(\mathbf{x}_i, y_i)\}_{i=1}^n$, where $\mathbf{x}_i \in \mathcal{R}^d$ is a benign example, $y_i \in \{1, \ldots, C\}$ is ground truth label often encoded as a one-hot vector $\mathbf{y}_i \in \{0,1\}^C$, and $C$ is the total number of classes, PGD-AT (Madry et al., 2018) can be formulated as the following min-max optimization problem:

$$\min_{\theta} \sum_{i=1}^n \max_{\mathbf{x}_i' \in \mathcal{S}(\mathbf{x}_i)} \ell(f_\theta(\mathbf{x}_i'), \mathbf{y}_i) \tag{1}$$

where $f_\theta$ is a model with parameter $\theta$. $\ell$ is the cross-entropy loss function, and $\mathcal{S}(\mathbf{x}) = \{\mathbf{x}' : ||\mathbf{x}' - \mathbf{x}||_p \leq \epsilon\}$ is an adversarial region centered at $\mathbf{x}$ with radius $\epsilon > 0$ under $l_p$-norm threat model.

The adversarial example $\mathbf{x}_i'$ can be obtained by projected gradient descent (PGD) to approximate the inner maximization in adversarial training, which randomly initializes a point within $\mathcal{S}(\mathbf{x}_i)$ and iteratively updates the point for $K$ steps with:

$$\mathbf{x}_i^{t+1} = \Pi_{\mathcal{S}(\mathbf{x}_i)}(\mathbf{x}_i^t + \alpha \cdot \text{sign}(\nabla_{\mathbf{x}} \ell(f_\theta(\mathbf{x}_i^t), \mathbf{y}_i))) \tag{2}$$

where $\Pi(.)$ is the projection, $\alpha$ is the attack step size, $t$ denotes iteration count, and $\mathbf{x}_i' = \mathbf{x}_i^K$.

### 3.2 DISTRIBUTIONAL DIFFERENCE IN THE OUTPUTS OF ROBUST AND NON-ROBUST MODEL

The standard training approach encourages models to generate confident predictions regardless of the scenario, leading to overconfident outcomes when the testing distribution changes. In contrast, robust models, when compared to their standardly trained counterparts, possess superior calibration properties that exhibit low confidence in incorrect classified examples (Grabinski et al., 2022). In addition, a study conducted by Qin et al. (2021) has revealed that poorly calibrated examples are more vulnerable to attacks. The interplay between robustness and confidence calibration motivates us to enhance the information inherent in labels. Therefore, we initiate our analysis by examining the differences in output distribution between robust and normal models.

**Robust model generates a random output on OOD data** When there is a significant distribution shift in the testing data, a well-calibrated model is expected to display uncertainty in its predictions by assigning uniformly random probabilities to unseen examples. To analyze the difference in output distributions when predicting out-of-distribution (OOD) data, we follow the approach by Snoek et al. (2019); Qin et al. (2021). Specifically, we compare the histogram of the output entropy of the models. We use ResNet-18 to train the models on the CIFAR-10 (in-distribution) training set and evaluate the CIFAR-100 (OOD) testing set. Given that most categories in CIFAR-100 were not present during training, we expect the models to reveal suspicion. As shown in Figure 1, the non-robust model has low entropy (high confidence) on OOD data, while the robust model exhibits high uncertainty on average.

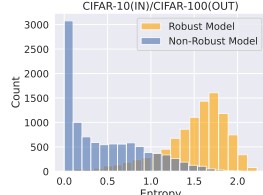

Figure 1: Output distribution on OOD data. Both models are trained on CIFAR-10 and tested on CIFAR-100.

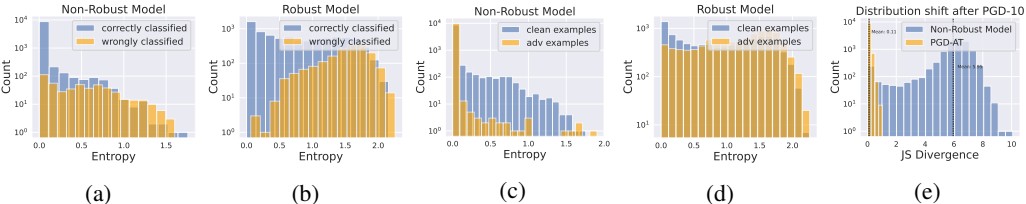

Figure 2: (a) and (b) are entropy distributions on the correctly classified and misclassified examples on the standard and robust model respectively. (c) and (d) are entropy distributions on the clean and adversarial examples on the standard and robust model respectively. (e) shows histograms of JS divergence for output distribution shift under the PGD-10 attack.

**Robust models are uncertain on incorrectly classified examples**  To investigate potential distributional differences in the behavior of standard and robust models when encountering correctly or incorrectly classified examples, we consider the model's confidence level in its predictions. Lower confidence levels are typically associated with higher error rates if the model is well-calibrated. To this end, we experiment by using a ResNet-18 model trained and evaluated on the CIFAR-10 dataset to demonstrate the distributional differences between correctly and incorrectly classified examples for both standard and robust models. Specifically, Figure 2a illustrates that the standard model exhibits low entropy levels for correctly classified examples, but relatively uniform entropy levels for incorrectly classified ones. Higher confidence in the prediction does not guarantee better performance. On the other hand, Figure 2b shows that the robust model tends to exhibit relatively high entropy levels (low confidence) for misclassified examples. We can infer that when the robust model is confident in its prediction, the classification accuracy is likely to be high.

**Output distribution of models on clean or adversarial examples are consistent**  Several prior studies (Zhao et al., 2022; Cui et al., 2021) have suggested learning clean images' representation from the standard model and adversarial example's representation from the robust model to improve robustness while maintaining accuracy. However, the underlying assumption that the robust model exhibits comparable representations of clean images to those generated by the standard model has not been thoroughly examined. Therefore, we investigate whether these models show comparable behavior when presented with clean and adversarial (PGD-10) examples on the CIFAR-10 dataset.

We demonstrate that the standard model exhibits low entropy in both scenarios (Figure 2c), whereas the robust model yields high entropy on average (Figure 2d). Additionally, Figure 2e reveals two models' histograms of JS divergence, representing the extent of output distribution shift when input is attacked. We can observe that even if robust models are attacked successfully, the change of output distribution measured in JS divergence is still small compared to standard models. The robust models show higher consistency (low JS divergence), while the standard models make drastic output changes. Therefore, promoting learning from standard models on normal examples may not be ideal for robust models, as robust models do not generate high confidence output on clean data.

## 4 METHODOLOGY

### 4.1 MOTIVATION: RECTIFY LABELS IN A NOISE-AWARE MANNER

Based on previous analysis, robust models should satisfy three key properties: first, they generate nearly random probability on OOD data; second, they demonstrate high uncertainty when it is likely to make a mistake; and third, they exhibit output consistency for both clean examples and their adversarial counterparts. However, the one-hot label used in AT does not provide sufficient guidance to reflect real-world distribution. Restricting the output to fix hard labels can be toxic to adversarial training (Dong et al., 2022a), as it causes the model to memorize the labels (Dong et al., 2022b) to minimize training loss, but at the expense of losing generalization ability on the testing set. In a recent study conducted by Paleka & Sanyal (2022), it was found that uniform label noise has a similar degree of adverse impact as worst-case data poisoning. They also provide empirical evidence that real-world noise is less harmful than uniform-label noise. Specifically, the noise introduced by human annotators poses a lower adversarial risk than uniform label noise. Therefore, designing a label-softening mechanism that gradually approaches the true underlying distribution instead of assigning uniform noise as label smoothing is essential to improve robustness. Building on these in-

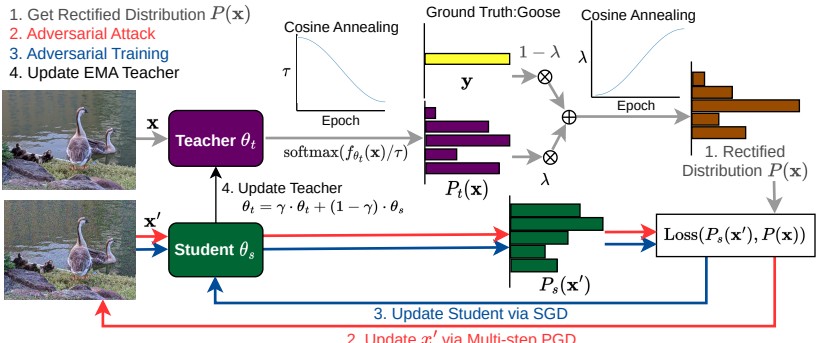

Figure 3: Overview of ADR.

sights, we propose a data-driven scheme, **A**nnealing Self-**D**istillation **R**ectification (ADR), to rectify labels in a noise-aware manner that mimics the behavior of human annotators.

## 4.2 ANNEALING SELF-DISTILLATION RECTIFICATION

To be specific, let $\theta_s$ represent the trained model's parameter to optimize, and $\theta_t$ be the EMA of $\theta_s$, which is updated by $\theta_t = \gamma \cdot \theta_t + (1 - \gamma) \cdot \theta_s$ where $\gamma$ is the decay factor. $P_t(\mathbf{x}_i)$ is EMA's output distribution on input $\mathbf{x}_i$, and $P_t(\mathbf{x}_i)^{(c)}$ serves as EMA's predicted probability for the class $c$. To obtain a rectified label, we first calculate the EMA's softened distribution $P_t(\mathbf{x}_i)$ with temperature $\tau$, which follows a cosine annealing from high to low. Since $f_{\theta_t}$ cannot provide sufficient knowledge at the beginning of training, the high temperature encourages the EMA's output to approach uniform noise. As $f_{\theta_t}$ becomes more accurate and robust, we anneal the temperature to make the distribution more descriptive of the inter-class relations. The smoothed distribution $P_t(\mathbf{x}_i)$ for $f_{\theta_t}$ is as follows,

$$P_t(\mathbf{x}_i) = \text{softmax}(f_{\theta_t}(\mathbf{x}_i)/\tau) \tag{3}$$

However, $f_{\theta_t}$ does not always classify correctly, especially when training is insufficient. To ensure the correct class has the highest probability so the target is unchanged, we interpolate the predicted distribution of $f_{\theta_t}$, $P_t(\mathbf{x}_i)$, with ground-truth one-hot $\mathbf{y}_i$, which is built from $y_i$, by an interpolation ratio $\lambda$. $\lambda$ follows an increasing cosine schedule, allowing us to trust EMA more over time. For each $\mathbf{x}_i$, we also adjust $\lambda$ to $\lambda_i$ to ensure the true class has the highest probability across the distribution.

$$\lambda_i = \text{clip}_{[0,1]}(\lambda - (P_t(\mathbf{x}_i)^{(\psi_i)} - P_t(\mathbf{x}_i)^{(y_i)})) \tag{4}$$

where $\psi_i$ exhibits the EMA's predicted class and $y_i$ represents the ground truth class. When EMA makes a correct prediction, that is $P_t(\mathbf{x}_i)^{(\psi_i)} = P_t(\mathbf{x}_i)^{(y_i)}$, there is no need to adjust the interpolation rate, otherwise, we decrease $\lambda$ by the amount that the EMA model makes mistake on $\mathbf{x}_i$ and then clip to the $[0, 1]$ range. Finally, the rectified distribution $P(\mathbf{x}_i)$ used for adversarial attack and training is carried out as

$$P(\mathbf{x}_i) = \lambda_i \cdot P_t(\mathbf{x}_i) + (1 - \lambda_i) \cdot \mathbf{y}_i \tag{5}$$

We use the rectified label $P(\mathbf{x}_i)$ to replace the ground truth label $\mathbf{y}_i$ in Equation 1 and Equation 2 to conduct adversarial training. Similarly, the softened $P(\mathbf{x}_i)$ can be applied in other adversarial training algorithms, e.g. TRADES (Zhang et al., 2019), by replacing the hard labels. We illustrate the overview of ADR in Figure 3 and summarize the pseudo-code in Appendix B.

## 5 EXPERIMENTS

In this section, we compare the proposed ADR to PGD-AT and TRADES in Table 1. We further investigate the efficacy of ADR in conjunction with model-weight-space smoothing techniques Weight Average (WA) (Izmailov et al., 2018; Gowal et al., 2020) and Adversarial Weight Perturbation (AWP) (Wu et al., 2020) with ResNet18 (He et al., 2016a) in Table 2. Experiments are conducted on well-established benchmark datasets, including CIFAR-10, CIFAR-100 (Krizhevsky et al., 2009), and TinyImageNet-200 (Le & Yang, 2015; Deng et al., 2009). We highlight that

Table 1: Test accuracy (%) of ADR compared to PGD-AT and TRADES with ResNet18. *Best* refers to the checkpoint with the highest robust accuracy on the evaluation set under PGD-10 evaluation. *Final* is the last checkpoint, and *Diff* is the difference of accuracy between *Best* and *Final*. The best results and the smallest performance differences are marked in **bold**.

| Dataset | Method | AutoAttack(%) | | | Standard Acc.(%) | | |
|---|---|---|---|---|---|---|---|
| | | Best | Final | Diff. | Best | Final | Diff. |
| CIFAR-10 | AT | 48.81 | 43.19 | 5.62 | **82.52** | 83.77 | **-1.25** |
| | AT + ADR | **50.38** | **47.18** | **3.20** | 82.41 | **85.08** | -2.67 |
| | TRADES | 50.1 | 48.17 | 1.93 | 82.95 | 82.42 | -0.53 |
| | TRADES + ADR | **51.02** | **50.4** | **0.62** | **83.4** | **83.76** | **-0.36** |
| CIFAR-100 | AT | 24.95 | 19.66 | 5.29 | 55.81 | 56.58 | **-0.77** |
| | AT + ADR | **26.87** | **24.23** | **2.64** | **56.1** | **56.95** | -0.85 |
| | TRADES | 24.71 | 23.78 | **0.93** | 56.37 | **56.01** | **0.36** |
| | TRADES + ADR | **26.42** | **25.15** | 1.27 | **56.54** | 54.42 | 2.12 |
| TinyImageNet-200 | AT | 18.06 | 15.86 | 2.2 | 45.87 | **49.35** | -3.48 |
| | AT + ADR | **19.46** | **18.83** | **0.63** | **48.19** | 47.65 | **0.54** |
| | TRADES | 17.35 | 16.4 | 0.95 | 48.49 | 47.62 | **0.87** |
| | TRADES + ADR | **19.17** | **18.86** | **0.31** | **51.82** | **50.87** | 0.95 |

defending against attacks on datasets with enormous classes is more difficult as the model's decision boundary becomes complex. We also provide experiments on Wide ResNet (WRN-34-10) (Zagoruyko & Komodakis, 2016) with additional data compared to other state-of-the-art methods reported on RobustBench (Croce et al., 2021) in Table 3. Our findings reveal that ADR outperforms methods across different architectures, irrespective of the availability of additional data.

## 5.1 TRAINING AND EVALUATION SETUP

We perform adversarial training with perturbation budget $\epsilon = 8/255$ under $l_\infty$-norm in all experiments. In training, we use the 10-step PGD adversary with step size $\alpha = 2/255$. We adopt $\beta = 6$ for TRADES as outlined in the original paper. The models are trained using the SGD optimizer with Nesterov momentum of 0.9, weight decay 0.0005, and a batch size of 128. The initial learning rate is set to 0.1 and divided by 10 at 50% and 75% of the total training epochs. Simple data augmentations include $32 \times 32$ random crop with 4-pixel padding and random horizontal flip (Rice et al., 2020; Gowal et al., 2020; Pang et al., 2021) are applied in all experiments. Following Wu et al. (2020); Gowal et al. (2020), we choose radius 0.005 for AWP and decay rate $\gamma = 0.995$ for WA. For CIFAR-10/100, we use 200 total training epochs, $\lambda$ follows cosine scheduling from 0.7 to 0.95, and $\tau$ is annealed with cosine decreasing from 2.5 to 2 on CIFAR-10 and 1.5 to 1 on CIFAR-100, respectively. As for TinyImageNet-200, we crop the image size to $64 \times 64$ and use 80 training epochs. We adjust $\lambda$ from 0.5 to 0.9 and $\tau$ from 2 to 1.5 on this dataset.

During training, we evaluate the model with PGD-10 and select the model that has the highest robust accuracy on the validation set with early stopping (Rice et al., 2020). For testing, we use AutoAttack (Croce & Hein, 2020b) which comprises an ensemble of 4 attacks including APGD-CE (Croce & Hein, 2020b), APGD-DLR (Croce & Hein, 2020b), FAB (Croce & Hein, 2020a) and Square attack (Andriushchenko et al., 2020) for rigorous evaluation. To eliminate the possibility of gradient obfuscations (Athalye et al., 2018), we provide sanity checks in Appendix E. Unless otherwise specified, the robust accuracy (RA) is computed under AutoAttack to demonstrate the model's generalization ability on unseen attacks. The computation cost analysis is attached in Appendix G.

## 5.2 SUPERIOR PERFORMANCE ACROSS ROBUSTIFIED METHODS AND DATASETS

Table 1 demonstrates the results of ADR combined with PGD-AT and TRADES on CIFAR-10, CIFAR-100, and TinyImageNet-200. Our initial observations reveal that robust overfitting exists in all baselines, with differences between final and best early-stopping robust accuracies as large as 5.62% on CIFAR-10 while the standard accuracy (SA) remains stable with more training epochs.

Table 2: Test accuracy (%) of ADR combining with WA and AWP. The best results are marked in **bold**. The performance improvements and degradation are reported in red and blue numbers.

| Method | ResNet-18 | | WRN-34-10 | |
|---|---|---|---|---|
| | AutoAttack | Standard Acc. | AutoAttack | Standard Acc. |
| AT | 48.81 | 82.52 | 52.12 | 85.15 |
| + WA | 49.93 (+ 1.12) | **83.71** (+ 1.19) | 53.97 (+ 1.85) | 83.48 (- 1.67) |
| + WA + AWP | 50.72 (+ 1.91) | 82.91 (+ 0.39) | 55.01 (+ 2.89) | **87.42** (+ 2.27) |
| + ADR | 50.38 (+ 1.57) | 82.41 (- 0.11) | 53.25 (+ 1.13) | 84.67 (- 0.48) |
| + WA + ADR | 50.85 (+ 2.04) | 82.89 (+ 0.37) | 54.10 (+ 1.98) | 82.93 (- 2.22) |
| + WA + AWP + ADR | **51.18** (+ 2.37) | 83.26 (+ 0.74) | **55.26** (+ 3.14) | 86.11 (+ 0.96) |

(a) CIFAR-10

| Method | ResNet-18 | | WRN-34-10 | |
|---|---|---|---|---|
| | AutoAttack | Standard Acc. | AutoAttack | Standard Acc. |
| AT | 24.95 | 55.81 | 28.45 | 61.12 |
| + WA | 26.27 (+ 1.32) | 53.54 (- 2.27) | 30.22 (+ 1.77) | 60.04 (- 1.08) |
| + WA + AWP | 27.36 (+ 2.41) | **59.06** (+ 3.25) | 30.73 (+ 2.28) | **63.11** (+ 1.99) |
| + ADR | 26.87 (+ 1.92) | 56.10 (+ 0.29) | 29.35 (+ 0.90) | 59.76 (- 1.36) |
| + WA + ADR | 27.51 (+ 2.56) | 58.30 (+ 2.49) | 30.46 (+ 2.01) | 57.42 (- 3.70) |
| + WA + AWP + ADR | **28.50** (+ 3.55) | 57.36 (+ 1.55) | **31.60** (+ 3.15) | 62.21 (+ 1.09) |

(b) CIFAR-100

| Method | ResNet-18 | | WRN-34-10 | |
|---|---|---|---|---|
| | AutoAttack | Standard Acc. | AutoAttack | Standard Acc. |
| AT | 18.06 | 45.87 | 20.76 | 49.11 |
| + WA | 19.30 (+ 1.24) | **49.10** (+ 3.23) | 22.77 (+ 2.01) | 53.21 (+ 4.10) |
| + WA + AWP | 19.58 (+ 1.52) | 48.61 (+ 2.74) | 23.22 (+ 2.46) | **53.35** (+ 4.42) |
| + ADR | 19.46 (+ 1.40) | 48.19 (+ 2.32) | 21.85 (+ 1.09) | 51.52 (+ 2.41) |
| + WA + ADR | **20.23** (+ 2.17) | 48.55 (+ 2.68) | 23.01 (+ 2.25) | 51.03 (+ 1.92) |
| + WA + AWP + ADR | 20.12 (+ 2.06) | 48.27 (+ 2.40) | **23.33** (+ 2.57) | 51.44 (+ 2.33) |

(c) TinyImageNet-200

When combined with our approach, we observe consistent improvements across datasets, with reduced robust overfitting gaps from 5.62% to 3.2% on CIFAR-10, 5.29% to 2.64% on CIFAR-100, and 2.2% to 0.63% on TinyImageNet-200. Furthermore, the robust accuracy improves by 0.92% to 1.92% across experiments. By alleviating robust overfitting, the best checkpoints are closer to the end of the training, thereby improving the SA in most settings. Our findings indicate that ADR can effectively enhance the RA-SA trade-off by improving robustness while achieving higher standard accuracy. We also present performance variation across multiple reruns in Appendix H.

## 5.3 COMBING WITH WEIGHT SPACE SMOOTHING TECHNIQUES AND LARGER ARCHITECTURE

The proposed ADR can be integrated with other AT techniques to boost robustness further (Table 2). Additional experiments with TRADES are presented in Appendix C. WA (Gowal et al., 2020) and AWP (Wu et al., 2020) are the model-weight-space smoothing techniques that improve the stability and performance of AT. In our case, we can acquire WA result by evaluating $\theta_t$ as it maintains the EMA of the trained model's weight. Combining ADR with AWP and WA, we obtain large gains in RA ranging from 1.12% to 3.55% and 0.37% to 3.23% in SA compared to the ResNet-18 baselines.

Following prior works (Chen et al., 2021; Addepalli et al., 2022), we additionally use Wide ResNet (WRN-34-10) to demonstrate that ADR scales to larger architectures and improves RA and SA. The result shows that our method effectively enhances robust accuracy up to 3.14% on CIFAR-10, 3.15% on CIFAR-100, and 2.57% on TinyImageNet-200 compared to each of its baselines. Notably, we use the same $\lambda$, $\tau$ as ResNet-18 for WRN-34-10, which might not be optimal, to reduce the cost of hyperparameter searching. Therefore, we observe a slight drop in standard accuracy in some WRN cases. Nevertheless, ADR still outperforms baselines in robustness without tuning hyper-parameters.

Table 3: Comparison of ADR with related works on CIFAR-100.

| Architecture | Method | Extra Data | AutoAttack. | Standard Acc. |
|---|---|---|---|---|
| ResNet18 | Zhang et al. (2022) | - | 26.03 | 58.17 |
| | Dong et al. (2022b) | - | 26.30 | 56.45 |
| | Dong et al. (2022a) | - | 26.36 | 58.80 |
| | Addepalli et al. (2022) | - | 27.62 | 66.69 |
| | AT+ADR (Ours) | - | **28.50** | 57.36 |
| Preact-ResNet18 | Rebuffi et al. (2021a) | DDPM | 28.50 | 56.87 |
| | Rade & Moosavi-Dezfooli (2022) | DDPM | 28.88 | 61.50 |
| | AT+ADR (Ours) | DDPM | **29.59** | 57.88 |
| WRN-34-10 | Chen & Lee (2021) | - | 30.59 | 64.07 |
| | Jia et al. (2022) | - | 30.77 | 64.89 |
| | Sehwag et al. (2022) | DDPM | 31.15 | 65.93 |
| | Cui et al. (2021) | - | 31.20 | 62.99 |
| | Addepalli et al. (2022) | - | 31.30 | 68.74 |
| | AT+ADR (Ours) | - | 31.60 | 62.21 |
| | AT+ADR (Ours) | DDPM | **32.19** | 59.60 |

## 5.4 COMPARISON WITH RELATED WORKS AND USE ADDITIONAL DATA ON CIFAR-100

Table 3 compares our proposed ADR defense against related works on a more challenging CIFAR-100 dataset. We select leading methods (Rade & Moosavi-Dezfooli, 2022; Gowal et al., 2020; Rebuffi et al., 2021a; Addepalli et al., 2022; Cui et al., 2021; Sehwag et al., 2022; Jia et al., 2022; Chen & Lee, 2021) on RobustBench and methods similar to ours which introduce smoothing in training labels (Dong et al., 2022b;a; Zhang et al., 2022) to make a fair comparison. Since knowledge distillation also promotes learning from the soft target, we discuss the benefits of ADR over those methods in Appendix I. The reported numbers are listed in their original papers or on RobustBench. We also provide a similar comparison on TinyImageNet-200 in Appendix D. Given the observed benefits of incorporating additional training data to promote robust generalization (Schmidt et al., 2018), we employ a DDPM (Ho et al., 2020) synthetic dataset (Gowal et al., 2020; Rebuffi et al., 2021a) composed of 1 million samples. Detailed experiment setup can be found in Appendix F.

Our experimentation with AT-WA-AWP-ADR on ResNet-18 yields a robust accuracy of 28.5% and a standard accuracy of 57.36%, comparable to Rebuffi et al. (2021a) that utilizes an additional 1M DDPM data on Preact-ResNet18, which yields an RA of 28.5% and SA of 56.87%. Remarkably, our model attains equal robustness and superior standard accuracy without using additional data when employing a similar-sized model. Similarly, we achieve an RA of 31.6% on WRN-34-10, while Sehwag et al. (2022) scores only 31.15% with additional data. Additionally, adding DDPM data in the training set leads to further improvement in robust accuracy for ADR, by 1.09% and 0.59% for ResNet-18 and WRN-34-10, respectively. In both cases, ADR achieves new state-of-the-art performance, both with and without additional data, on the CIFAR-100 benchmark. It is worth noting that some methods introduce extra auxiliary examples when training (Dong et al., 2022b; Rade & Moosavi-Dezfooli, 2022), and some bring complex augmentation into AT (Rade & Moosavi-Dezfooli, 2022; Rebuffi et al., 2021a), and so might obtain superior SA compared to ADR. However, regarding the optimal robustness to achieve, our empirical findings provide compelling evidence that rectifying training labels with a realistic distribution is a valuable approach.

## 5.5 ACHIEVING FLATTER WEIGHT LOSS LANDSCAPE

Several studies (Wu et al., 2020; Stutz et al., 2021) have found that a flatter weight loss landscape leads to a smaller robust generalization gap when the training process is sufficient. Many methods (Wu et al., 2020; Chen et al., 2021; Gowal et al., 2020; Zhang et al., 2019) addressing robust overfitting issues predominantly find flatter minima. We visualize the weight loss landscape by plotting the loss change when moving the weight $w$ along a random direction $d$ with magnitude $\alpha$. The direction $d$ is sampled from Gaussian distribution with filter normalization (Li et al., 2018). For each perturbed model, we generate adversarial examples on the fly with PGD-10 and calculate the mean loss across the testing set.

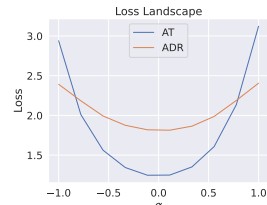

Figure 4: Model weight loss landscape comparison for AT and ADR.

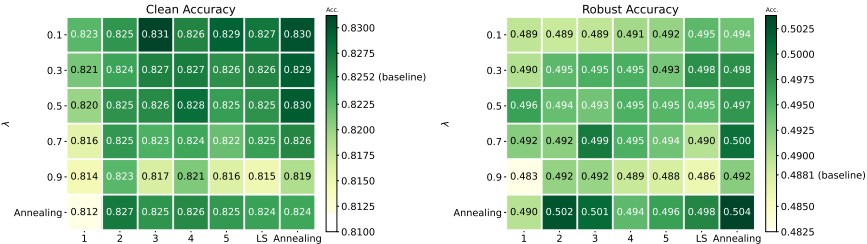

Figure 5: Effectiveness of different temperature $\tau$ and label interpolation factor $\lambda$ of ADR.

Figure 4 compares the weight loss landscape between AT and ADR on CIFAR-10. ADR achieves a flatter landscape, implying better robust generalization ability. While we smooth the label for ADR in training, we use the one-hot label as ground truth to calculate the cross-entropy loss in this experiment, so the model trained by ADR has a higher loss value than AT on average. Additionally, we visualize the loss landscape around the data point in Appendix E and observe a similar phenomenon that ADR produces a flatter loss landscape.

## 5.6 ACCURACY VS. ROBUSTNESS TRADE-OFF

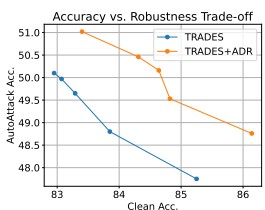

Mitigating the trade-off between accuracy and robustness has been challenging in the realm of adversarial training. To investigate if ADR is capable of reducing such trade-off, we combine ADR with TRADES and adjust the trade-off parameter $\beta$ to demonstrate the performance difference in terms of different values of $\beta$. The result is represented in Figure 6. We decrease $\beta$ from left to right, where a higher value of $\beta$ gives better robustness. We can clearly observe that TRADES+ADR achieves a better trade-off compared to that of using the hard label alone.

Figure 6: Comparison of the trade-off for ADR with TRADES.

## 5.7 ABLATION STUDY ON EFFECTIVENESS OF TEMPERATURE AND INTERPOLATION FACTOR

To disclose the impact of temperature $\tau$ and interpolation factor $\lambda$ on the proposed ADR, we conduct ablation studies to exhibit grid search outcomes of SA and RA in Figure 5. Throughout experiments, $\tau$ and $\lambda$ are held constant unless explicitly specified as "Annealing." to scrutinize the effects of varying parameter values. Furthermore, we include Label Smoothing (LS) in this study, which can be viewed as a special case where $\tau$ approaches infinity, to evaluate how data-driven smoothness improves performance. Our analysis reveals that in terms of clean accuracy, choosing a smaller $\lambda$ and suitable temperature $\tau = 3$ can achieve the best performance. As for the robustness, using moderately large $\lambda$ with appropriate $\tau$ can ensure that the training labels inherent enough inter-class relationship. Therefore, selecting a proper value of parameters is vital to maintaining clean accuracy while enhancing robustness. Our experiment also reveals that annealing in both temperature and interpolation factors is beneficial to improve robustness, which shows the efficacy of gradually increasing reliance on the EMA model.

## 6 CONCLUSION

In this paper, we characterize the key properties that distinguish robust and non-robust model output. We find that a robust model should exhibit good calibration and maintain output consistency on clean data and its adversarial counterpart. Based on this observation, we propose a data-driven label softening scheme ADR without the need for pre-trained resources or extensive computation overhead. To achieve this, we utilize the self-distillation EMA model to provide labeling guidance for the trained model, with increasing trust placed in the EMA as training progresses. Comprehensive experiments demonstrate that ADR effectively improves robustness, alleviates robust overfitting, and obtains a better trade-off in terms of accuracy and robustness. However, we note that the algorithm's optimal temperature and interpolation ratio depend on the dataset, and improper selection of these parameters can limit performance improvements. The automatic determination of optimal parameters in training will be an important future research direction that can further boost the robustness.

ETHICS STATEMENT

Adversarial training has the potential to improve the security, and reliability of machine learning systems. In practical settings, adversarial attacks can be employed by malevolent actors in an attempt to deceive machine learning systems. This phenomenon can engender grave consequences for domains such as autonomous driving vehicles and facial recognition. To enhance the security and reliability of machine learning systems, adversarial training can be employed to produce more dependable models. However, it is worth noting that robust models can also be exploited by ill-intentioned users. In the context of the CAPTCHA, adversarial perturbations can be added to images to distinguish between humans and robots since the robots are expected to be fooled by the adversarial data. If robots can attain robust models, they would not be susceptible to adversarial examples and could supply accurate answers. The advancement of model robustness may inspire people to formulate a better strategy to differentiate between humans and robots.

REPRODUCIBILITY

We describe the detailed experiment settings and hyperparameters in section 5.1 and Appendix F. Furthermore, the source code can be found in the supplementary materials to ensure the reproducibility of this project.

ACKKNOWLEDGEMENT

This work was supported in part by the National Science and Technology Council under Grants MOST 110-2634-F002-051, MOST 110-2222-E-002-014-MY3, NSTC 113-2923-E-002-010-MY2, NSTC-112-2634-F-002-002-MBK, by National Taiwan University under Grant NTU-CC-112L891006, and by Center of Data Intelligence: Technologies, Applications, and Systems under Grant NTU-112L900903.

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

## A  ADDITIONAL RELATED WORK: MITIGATE ROBUST OVERFITTING

The phenomenon of robust overfitting Rice et al. (2020) represents a significant challenge in AT, motivating researchers to explore various avenues for mitigation. One such approach is to use heuristic-driven augmentations Rebuffi et al. (2021b), such as CutMix Yun et al. (2019); Rebuffi et al. (2021a), DAJAT Addepalli et al. (2022), and CropShift Li & Spratling (2023), which employ sets of augmentations carefully designed to increase data diversity and alleviate robust overfitting. Another strategy involves the expansion of the training set, which offers a direct means to address overfitting. By incorporating additional unlabeled data Carmon et al. (2019) or high-quality generated images via deep diffusion probabilistic models (DDPM) Rebuffi et al. (2021a); Gowal et al. (2020); Sehwag et al. (2022), the introduction of an extra set of 500K pseudo-labeled images from 80M-TI Torralba et al. (2008) eliminates the occurrence of robust overfitting Rebuffi et al. (2021a). Despite the demonstrated effectiveness of extra data, increasing the size of the training set is computationally expensive, rendering AT infeasible for larger datasets.

Early stopping is a straightforward method for producing robust models Rice et al. (2020). However, due to the fact that the checkpoint of optimal robust accuracy and that of standard accuracy frequently do not align Chen et al. (2021), utilizing either of these measures can result in a compromise of overall performance. Weight Average (WA) Izmailov et al. (2018); Chen et al. (2021); Gowal et al. (2020) tracks the exponential moving average of model weights, thereby promoting flatter minima and increased robustness Hein & Andriushchenko (2017). Another effective regularization technique is Adversarial Weight Perturbation (AWP) Wu et al. (2020), which serves to promote flatness within the weight loss landscape and yields enhanced generalization capabilities Stutz et al. (2021).

## B  ALGORITHM FOR ADR

---

**Algorithm 1** **A**nnealing Self-**D**istillation **R**ectification (ADR)

---

**Input:** Training set $\mathcal{D} = \{(\mathbf{x}_i, y_i)\}_{i=1}^{n}$
**Parameter:** A classifier $f(.)$ with learnable parameters $\theta_s$; $\theta_t$ is exponential moving average of $\theta_s$ with decay rate $\gamma$; Batch size $m$; Learning rate $\eta$; Total training iterations $E$; Attack radius $\epsilon$, attack step size $\alpha$, number of attack iteration $K$; Temperature $\tau$; Interpolation ratio $\lambda$.

1: Randomly initialize the network parameters $\theta_s$, $\theta_t \leftarrow \theta_s$
2: **for** $e = 1$ to $E$ **do**
3:      Calculate $\tau_e$ according to the current iterations.
4:      Calculate $\lambda_e$ according to the current iterations.
5:      Sample a mini-batch $\{(\mathbf{x}_j, y_j)\}_{j=1}^{m}$ from $\mathcal{D}$
6:      **for** $j = 1$ to $m$ (in parallel) **do**
7:          $P_t(\mathbf{x}_j) \leftarrow \text{softmax}(f_{\theta_t}(\mathbf{x}_j)/\tau_e)$                 ▷ Calculate rectified label
8:          $\lambda_j = \text{clip}_{[0,1]}(\lambda_e - (P_t(\mathbf{x}_j)^{(\psi_j)} - P_t(\mathbf{x}_j)^{(y_j)}))$
9:          $P(\mathbf{x}_j) \leftarrow \lambda_j \cdot P_t(\mathbf{x}_j) + (1 - \lambda_j) \cdot \mathbf{y}_j$
10:          $\mathbf{x}'_j \leftarrow \mathbf{x}'_j + \epsilon \cdot \delta$, where $\delta \sim \text{Uniform}(-1, 1)$       ▷ Construct adversarial example
11:          **for** $k = 1$ to $K$ **do**
12:              $\mathbf{x}'_j = \Pi_{\mathcal{S}(\mathbf{x}'_j)}(\mathbf{x}'_j + \alpha \cdot \text{sign}(\nabla_{\mathbf{x}}\ell(f_{\theta_s}(\mathbf{x}'_j), P(\mathbf{x}_j))))$
13:          **end for**
14:      **end for**
15:      $\theta_s \leftarrow \theta_s - \frac{\eta}{m} \cdot \sum_{j=1}^{m} \nabla_{\theta_s}(\ell(f_{\theta_s}(\mathbf{x}'_j), P(\mathbf{x}_j)))$      ▷ Update model parameters
16:      $\theta_t \leftarrow \gamma \cdot \theta_t + (1 - \gamma) \cdot \theta_s$
17: **end for**

---

## C  TEST ACCURACY OF TRADES + ADR COMBING WITH WA AND AWP

In this study, we investigate the impact of combining TRADES and ADR with other adversarial training techniques, namely WA and AWP, on ResNet-18, as outlined in Table-4. Our experimental results demonstrate that leveraging a soft target generated by ADR yields exceptional robustness and

Table 4: Test accuracy (%) of ADR + TRADES combining with WA and AWP on ResNet-18. The best results are marked in **bold**. Robust Accuracy (RA) is evaluated with AutoAttack and Standard Accuracy (SA) refers to the accuracy of normal data.

| Method | CIFAR-10 | | CIFAR-100 | | TinyImageNet-200 | |
|---|---|---|---|---|---|---|
| | RA | SA | RA | SA | RA | SA |
| TRADES | 50.10 | 82.95 | 24.71 | 56.37 | 17.35 | 48.49 |
| + WA | 51.10 | 81.77 | 25.61 | 57.93 | 17.69 | 49.51 |
| + WA + AWP | 51.25 | 81.48 | 26.54 | 58.40 | 17.66 | 49.21 |
| + ADR | 51.02 | **83.40** | 26.42 | 56.54 | 19.17 | 51.82 |
| + WA + ADR | **51.28** | 82.69 | 27.11 | **58.58** | 19.17 | **51.99** |
| + WA + AWP + ADR | 50.59 | 80.84 | **27.63** | 57.16 | **19.48** | 51.38 |

Table 5: Comparison of ADR with other related works on TinyImageNet-200. The best result for each architecture is marked in **bold**

| Architecture | Method | AutoAttack. | Standard Acc. |
|---|---|---|---|
| ResNet-18 | AT | 18.06 | 45.87 |
| | Rade & Moosavi-Dezfooli (2022) | 18.14 | **52.60** |
| | Dong et al. (2022a) | 18.29 | 47.46 |
| | AT+ADR(Ours) | **20.23** | 48.55 |
| WRN-28-10 | Rebuffi et al. (2021b) | 21.83 | **53.27** |
| WRN-34-10 | AT | 20.76 | 49.11 |
| | AT+ADR(Ours) | **23.33** | 51.44 |

standard accuracy improvement, thereby achieving a superior trade-off. Specifically, ADR results in 1.18%, 2.92%, and 2.13% RA improvement and 0.45%, 2.21%, and 3.5% SA improvement on the baseline performance of CIFAR-10, CIFAR-100, and TinyImageNet-200, respectively.

However, we observe that the robustness improvement saturates or even slightly deteriorates when combining TRADES+ADR with weight smoothing techniques on CIFAR-10. This is attributed to the fact that TRADES already promotes learning and attacking adversarial data on a softened target, making the additional soft objective by ADR less effective. Nevertheless, the TRADES+ADR approach remains beneficial when dealing with more challenging datasets that contain a greater number of target classes combined with WA and AWP.

## D TEST ACCURACY (%) COMPARED WITH RELATED WORKS ON TINYIMAGENET-200.

We present ADR evaluated against related works Madry et al. (2018); Rade & Moosavi-Dezfooli (2022); Dong et al. (2022a); Rebuffi et al. (2021b) in Table-5 on the TinyImageNet-200 dataset, which is a more challenging robustness benchmark than CIFAR-10 or CIFAR-100 due to its larger class size and higher-resolution images, using the original numbers from their respective papers. A model trained on a larger class dataset often results in more complex decision boundaries, which increases the likelihood of an attacker identifying vulnerabilities in the model. Our experimental results demonstrate that ADR achieves state-of-the-art performance, improving RA by 1.94% to 2.17% when using ResNet-18, and achieving a remarkable 2.57% improvement over the baseline on WRN-34-10. In summary, we observe that ADR stands out in the challenging multi-class scenario.

## E SANITY CHECK FOR GRADIENT OBFUSCATION

Athalye et al. (2018) argued that some defenses improving the robustness by obfuscated gradient, which is introduced intentionally through non-differentiable operations or unintentionally through

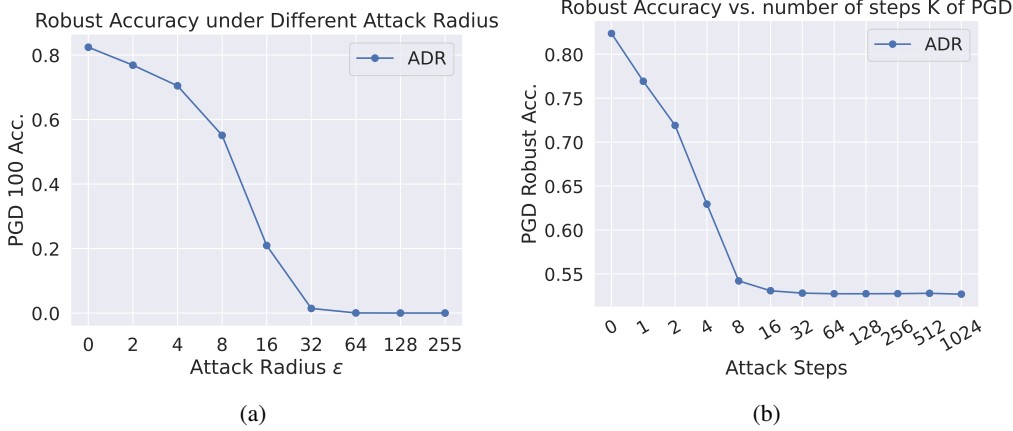

(a)

(b)

Figure 7: The changes of robust accuracy against different attack radius (7a) and attack steps (7b).

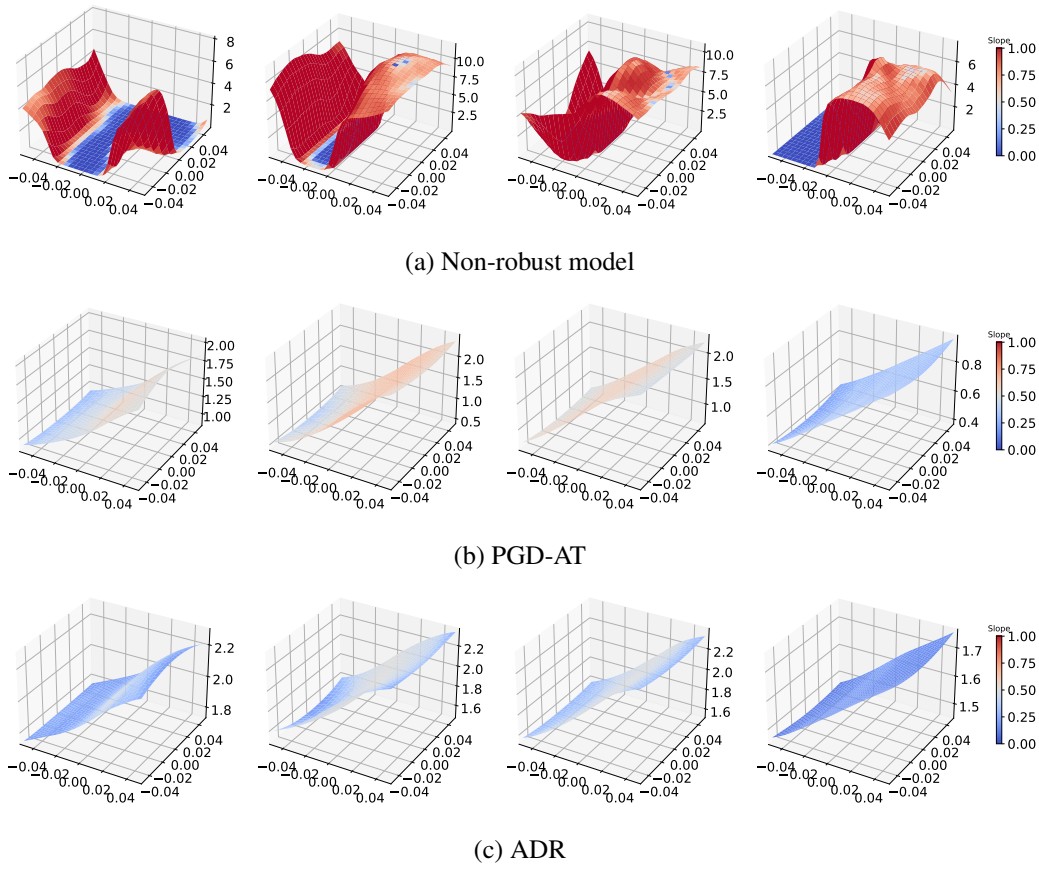

Figure 8: Comparison of loss landscapes for the non-robust model (the first row), PGD-AT model (the second row), and ADR model (the third row). Loss plots in each column are generated from the same image chosen from the CIFAR-10. Following the same setting as Engstrom et al. (2018); Chen et al. (2021), we plot the loss landscape function $z = loss(x \cdot r_1 + y \cdot r_2)$, where $r_1 = sign(\nabla_i f(i))$ and $r_2 \sim Rademacher(0.5)$. The $x$ and $y$ axes represent the magnitude of the perturbation added in each direction and the $z$ axis represents the loss. $i$ denotes the input and $f(.)$ is the trained model. The color represents the gradient magnitude of the loss landscape clipped in [0, 1] range, which conveys the smoothness of the surface.

numerical instability, can be circumvented. Though we already use AutoAttack Croce & Hein (2020b) which has been shown to provide a reliable estimation of robustness as an adversary throughout the experiments, we conduct additional evaluations with ADR to eliminate the possibility of gradient obfuscation. Following the guidelines by Carlini et al. (2019), we examine the impact of changes on the robustness with ResNet-18 trained by AT+ADR on CIFAR-10 against $l_\infty$ perturbation.

**Unbounded PGD attack** The unbounded PGD adversary should reduces model robustness to $0\%$. Figure 7a shows the changes of PGD-100 robust accuracy of AT+ADR with different attack radius $\epsilon$. The robust accuracy for AT+ADR monotonically drops to $0\%$ as the attack radius $\epsilon$ increases.

**Increasing PGD attack steps** Increasing the number of attack iterations should only marginally lower the robust accuracy. Figure 7b shows the changes in robust accuracy of AT+ADR with different steps of PGD attack. We can observe that the robust accuracy almost converges after $K = 16$, more steps of attack do not lead to lower robust accuracy.

**Inspect loss landscape around inputs** Figure 8 shows the loss landscape of Non-Robust, AT, and AT+ADR model around randomly selected examples from CIFAR-10. Compared to the Non-Robust model, the adversarially trained models (both PGD-AT and AT+ADR) have flattened the rugged landscape, which does not exhibit the typical patterns of gradient obfuscation Engstrom et al. (2018). It is notable that despite both PGD-AT and ADR having smooth landscapes, ADR has a lower gradient magnitude (dark blue color in the figure), which implies the loss changes for AT+ADR is smaller than AT when small perturbations is added to the input.

## F EXPERIMENT SETUP WHEN UTILIZING ADDITIONAL DATA

Following Gowal et al. (2020); Rebuffi et al. (2021a), we train Preact-ResNet18 (He et al., 2016b) and WRN-34-10 with SiLU (Hendrycks & Gimpel, 2016) as the activation function when utilizing synthetic data. We adopt a rigorous experimental design following (Gowal et al., 2020; Rebuffi et al., 2021a), training the Preact-ResNet18 (He et al., 2016b) and WRN-34-10 architectures with SiLU (Hendrycks & Gimpel, 2016) as the activation function when utilizing synthetic data. We leverage cyclic learning rates (Smith & Topin, 2017) with cosine annealing (Loshchilov & Hutter, 2017) by setting the maximum learning rate to $0.4$ and warmup period of 10 epochs, ultimately training for a total of $400$ CIFAR-100 equivalent epochs. Our training batch size is set to $1024$, with 75% of the batch composed of the synthetic data. We maintain consistency with other experiment details outlined in section 5.1.

## G COMPUTATION COST ANALYSIS

A standard 10 step AT includes 10 forwards and backward to find the worst-case perturbation when given a normal data point $\mathbf{x}$. After we generate the adversarial data $\mathbf{x}'$, it requires an additional 1 forward and backward pass to optimize the model. We will need 11 forward and backward pass per iteration to conduct PGD-10 adversarial training. When introducing ADR to rectify the targets, an additional forward is needed for the EMA model. We need 12 forward and 11 backward pass in total.

We provide time per epoch for adversarial training in Table-6. The experiment is reported by running each algorithm on a single NVIDIA RTX A6000 GPU with batch size 128. From the table, we can infer that the computation cost introduced by ADR is relatively small (1.83% on ResNet-18 4.45% on WRN-34-10 on average) while the overhead brought by AWP is a lot higher (12.6% on ResNet-18, 12.8% on WRN-34-10 on average). We can achieve similar robustness improvement to AWP with ADR with less computation cost required.

## H VARIANCE ACROSS RERUNS

Table-7 presents the results of five repeated runs for AT and the proposed defenses AT+ADR on CIFAR-10 with ResNet-18. Our findings indicate that the proposed ADR approach consistently

Table 6: Computational cost analysis for ADR combining with AT techniques on CIFAR-10, CIFAR-100 and TinyImageNet-200 with ResNet-18 and WRN-34-10. We report the time required per epoch in seconds tested on a single NVIDIA RTX A6000 GPU.

| Dataset | Architecture | Method | Time/epoch (sec) |
|---|---|---|---|
| CIFAR-10 | ResNet-18 | AT | 81 |
| | | +ADR | 83 |
| | | +AWP | 92 |
| | | +AWP+ADR | 93 |
| | WRN-34-10 | AT | 575 |
| | | +ADR | 610 |
| | | +AWP | 650 |
| | | +AWP+ADR | 695 |
| CIFAR-100 | ResNet-18 | AT | 81 |
| | | +ADR | 83 |
| | | +AWP | 91 |
| | | +AWP+ADR | 93 |
| | WRN-34-10 | AT | 586 |
| | | +ADR | 598 |
| | | +AWP | 661 |
| | | +AWP+ADR | 672 |
| TinyImageNet-200 | ResNet-18 | - | 584 |
| | | +ADR | 593 |
| | | +AWP | 654 |
| | | +AWP+ADR | 662 |
| | WRN-34-10 | AT | 4356 |
| | | +ADR | 4594 |
| | | +AWP | 4904 |
| | | +AWP+ADR | 5127 |

Table 7: Variation in performance (%) of AT and ADR on ResNet-18 across 5 reruns on CIFAR-10. The robust accuracy is evaluated under the PGD-100 attack.

| | AT | | AT+ADR | |
|---|---|---|---|---|
| | Robust Acc. | Standard Acc. | Robust Acc. | Standard Acc. |
| Run-1 | 52.80 | 82.52 | 55.13 | 82.41 |
| Run-2 | 52.28 | 82.41 | 54.88 | 82.21 |
| Run-3 | 52.74 | 82.39 | 54.92 | 82.18 |
| Run-4 | 52.55 | 82.30 | 54.91 | 82.68 |
| Run-5 | 52.63 | 82.31 | 54.73 | 82.31 |
| Average | 52.60 | 82.38 | 54.91 | 82.36 |
| Standard Deviation | 0.182 | 0.079 | 0.128 | 0.180 |

Table 8: Comparison of ADR with knowledge distillation methods on CIFAR-100 with ResNet-18.

| Method | AutoAttack | Standard Acc. |
|---|---|---|
| ARD (Goldblum et al., 2020b) | 25.65 | 60.64 |
| RSLAD (Zi et al., 2021) | 26.70 | 57.74 |
| IAD (Zhu et al., 2022) | 25.84 | 55.19 |
| MT (Zhang et al., 2022) | 26.03 | 58.10 |
| AT+ADR | 28.50 | 57.36 |

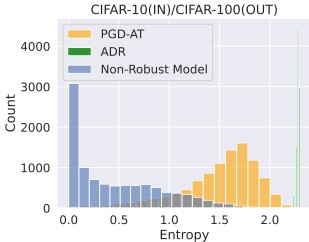
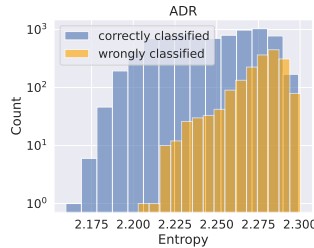
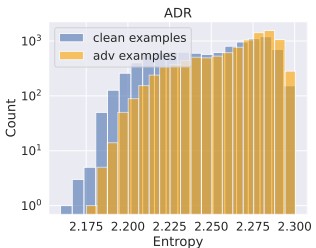

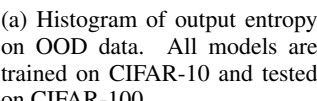

(a) Histogram of output entropy on OOD data. All models are trained on CIFAR-10 and tested on CIFAR-100.

(b) Histogram of entropy distributions on the correctly classified and misclassified examples for PGD-AT + ADR model.

(c) Histogram of entropy distributions on the clean and adversarial examples for PGD-AT + ADR model.

Figure 9: ADR model's output distribution exhibits similar properties as AT robust models

outperforms AT as evidenced by a higher mean (54.91% for AT+ADR compared to 52.6 for AT) and lower standard deviation in robust accuracy (0.128% for AT+ADR compared to 0.182% for AT). This suggests that ADR provides superior performance and greater stability in terms of robustness. While there is a larger standard deviation observed in standard accuracy when ADR is combined with AT, we consider the variance to be within an acceptable range (0.18%). Our results demonstrate that ADR yields stable performance that is independent of random seeds or other initialization states.

## I DISCUSSION ABOUT DIFFERENCE WITH KNOWLEDGE DISTILLATION BASED METHODS

Knowledge distillation in adversarial training generally requires pre-trained resources as teacher models, on the other hand, we do not acquire anything in advance when using ADR. Therefore, it is a great advantage that ADR can achieve satisfactory results without additional resources. Furthermore, in the phase of training a teacher model for knowledge distillation, the hard label still encourages the teacher to exhibit nearly one-hot output and may lead to overconfident results. It is challenging to control the softness of the target when distillate to the student model because the teacher might output nearly one-hot distribution for some examples that have high confidence and rather smooth distribution for others. The bias and confidence from the teacher model might be incorrectly inherited by the student model. Instead of forcing the model to learn from one-hot ground truth, the noise-aware ADR label promotes the model to adapt label noise from the early training stage. We can control the smoothness of the learning target by manipulating the $\lambda$ and $\tau$, as $\lambda$ controls the fraction of the ground truth class and $\tau$ decides the smoothness of the noise composed in the rectified label. We provide additional results in Table 8 to validate that ADR is superior to the knowledge distillation based methods.

## J LIMITATIONS

In this work, we proposed ADR that employs a self-distillate EMA model to generate a finely calibrated soft label to enhance the robustness of models against adversarial attacks. However, we observe that the optimal parameters for each dataset vary. Thus, selecting appropriate parameters that suit the current training state is crucial to ensure optimal performance. It is also noteworthy that while ADR demonstrates its efficacy in improving the performance of TRADES, the extent of improvement saturates when combined with other adversarial training techniques (WA, AWP) on fewer class datasets e.g. CIFAR-10. This outcome could be attributed to the fact that TRADES already promotes attacking and learning the data with soft targets generated by the trained model itself, and these additional techniques further smooth the model weight space. Thus, when the target class is fewer, the improvement provided by ADR, which also emphasizes a smooth objective, becomes indistinguishable when all techniques are employed simultaneously.

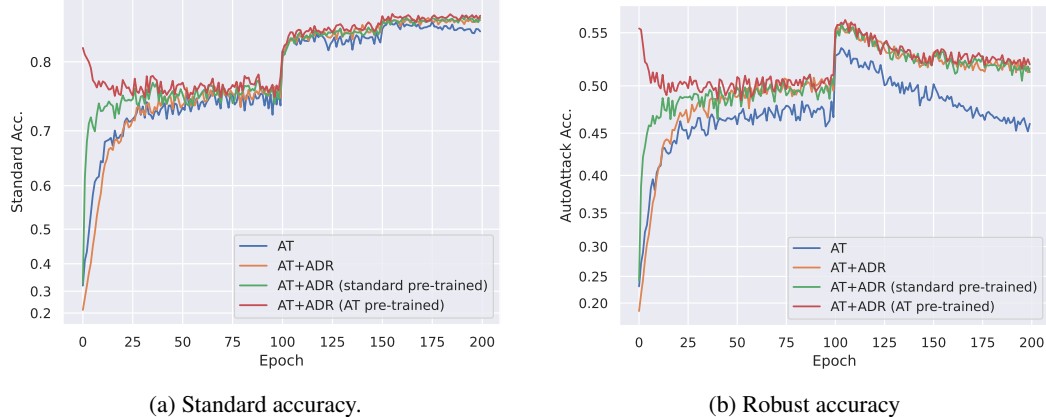

|                | (a) Standard accuracy. | (b) Robust accuracy |
|----------------|------------------------|---------------------|

Figure 10: The standard and robust accuracy training curve when transferring ADR from standard and adversarial pre-trained models.

|                                      | AutoAttack | Standard Acc. |
|--------------------------------------|------------|---------------|
| AT                                   | 48.81      | 82.52         |
| AT+ADR                               | 50.38      | 82.41         |
| AT+ADR + standard model initialized  | 49.45      | 82.22         |
| AT+ADR + AT model initialized        | 50.11      | 83.66         |

Table 9: Transferring ADR from pre-trained checkpoints as initialization. The standard model is trained with the normal classification objective, and the AT model is pre-trained with PGD-AT.

## K  A CLOSER LOOK AT ADR MODELS' OUTPUT DISTRIBUTION

From Figure 9a, we can see that the ADR model trained on CIFAR-10 is more uncertain than the model trained with PGD-AT when encountering out-of-distribution data (CIFAR-100). It supports our claim that the robust model should not be over-confident when seeing something it has never seen before. From Figure 9b and Figure 9c, we can observe that the ADR-trained model exhibits similar output distribution as the model trained with PGD-AT (Figure 2b and Figure 2d), except that the ADR-trained model exhibits higher entropy levels in general. The reason behind this phenomenon is that ADR builds a sofer target as training labels, so the output of the ADR model will also be smoother, resulting in higher entropy levels on average.

## L  ADR TRAINING TRANSFERRING FROM PRE-TRAINED INITIALIZATION

In this experiment, we leverage pre-trained models for the initialization of the ADR process. Specifically, we employ ResNet-18 on the CIFAR-10 dataset, adhering to all other parameter settings outlined in Section 5.1. The key variation lies in the initialization of the model with either a standard or PGD-AT pre-trained model, followed by subsequent ADR training. Our reporting in Table 9 is based on the optimal checkpoint obtained during our comprehensive evaluation. Moreover, to offer a more comprehensive view of the observed trends, we present line charts depicting the performance trajectories with both clean and robust accuracy in Figure 10.

The experimental findings reveal a consistent trend wherein ADR consistently outperforms the baseline in terms of robust accuracy, regardless of whether it is initialized with a pre-trained model or not. The utilization of either a standard model or a PGD-AT pre-trained weight as an initialization fails to further augment the maximum robustness achievable through ADR. Nevertheless, it is noteworthy that employing an AT pre-trained checkpoint results in a notable enhancement of standard accuracy by $1.25\%$ when compared to training with random initialization. This outcome underscores the potential for mitigating the accuracy-robustness tradeoff, indicating the feasibility of achieving improved performance by utilizing an AT pre-trained model during the initialization phase.

