# OpenReview forum: "Annealing Self-Distillation Rectification Improves Adversarial Training"
_ICLR.cc/2024/Conference — ICLR 2024 poster_

### Official Review · Reviewer_VvcZ · 2023-10-27

**Soundness:** 3 good
**Presentation:** 4 excellent
**Contribution:** 2 fair
**Rating:** 6
**Confidence:** 3

**Summary:**

The authors have proposed a self-distillation method to generate soft labels in AT. The soft labels are crafted by the natural logits of the teacher model and the ground-truth. They clip the weight of soft labels to keep the correct classes having the highest confidence. Besides, Annealing and temperature are introduced to adjust the labels adaptive. Empirical evaluation on  three benchmark datasets shows its improvement on robustness based on baselines.

**Strengths:**

1. A lot of experiments have been done to show its effectiveness on robustness compared with other knowledge distillation methods, which makes it credible.
2. The paper is well-organized and easy to understand.
3. The paper has shown abunbant experimental details, with good reproducibility.

**Weaknesses:**

1. The motivation is too easy and there seem no new insights in the manuscript.
2. The results on Table 2 show that on WRN on three datasets, AT+ADR has a weaker performance compared with AT+WA, and AT+WA+ADR also has a weaker performance compared with AT+WA+AWP. It may demonstrte ADR doesn't work well compared with WA and AWP on large models.
3. The comparison with RobustBench in the paper is not fair, results on AT+ADR rather than AT+WA+AWP+ADR should be used. In this comparison, ADR achieves atate-of-the-art robust performance on ResNet-18 but has a poor performance on WRN.

**Questions:**

Because it has little novelty, its performance is now the most significant evaluation indicator. Its performance on WRN is not competitive enough. I will increase my score if better results come out.

=========After rebuttal=============
The authors' response address most of my concerns. I thus am willing to increase the rating to 6

---

> ### Author Response · Authors · 2023-11-21
> **Response for Reviewer VvcZ**
>
> Dear reviewer VvcZ,
>
> Thank you for acknowledging our work's experimental result as credible and we are glad that you find it well-organized. We address your concerns as the following:
>
> **Q: The motivation is too easy and there seem no new insights in the manuscript.**
>
> We highlight that we observe the key property that makes the robust model reliable in Section 3.2, the robust model is well-calibrated and consistent with its output when input is perturbed, this motivates us to design a unified rectified label to enhance the efficacy of adversarial defense.
>
> In our observations, we discover that adversarially trained networks consistently exhibit output consistency between adversarial images and their clean counterparts. Some previous studies suggest the incorporation of dual targets during adversarial training, encouraging models to learn from both an adversarially trained model on adversarial images and a naturally trained model on normal images (Cui et al., 2021; Zhao et al., 2022). While seemingly intuitive in mitigating the trade-off between robustness and accuracy, our observations challenge the appropriateness of this approach. We contend that learning from two targets, one with high confidence from the normal model and the other with lower confidence from the adversarially trained model, is not conducive to achieving consistency for images with noise. Therefore, we propose composing a rectified label in ADR and urge that all images within the $l_p$ norm neighborhood of a given input should be targeted using the rectified label rather than the one-hot one. The observations in Section 3 motivate us to design ADR that effectively prevents the network from becoming over-confident and provides semantically aware smoothing rather than a uniform one. We believe that we do provide some useful insights for the community.
>
> **Q: The results in Table 2 show that on WRN on three datasets, AT+ADR has a weaker performance compared with AT+WA, and AT+WA+ADR also has a weaker performance compared with AT+WA+AWP. It may demonstrate ADR doesn't work well compared with WA and AWP on large models.**
>
> ADR with WA and AWP should be viewed as complementary rather than competing approaches. These methods target different aspects of enhancing adversarial training.
>
> In the realm of adversarial defense, we can encapsulate the directions of research aimed at enhancing the efficacy of adversarial training into the following key areas:
> * Different training objectives. e.g. PGD-AT (Madry et al., 2018), TRADES (Zhang et al., 2019).
> * Encourage a flatter model-weight landscape. e.g.  WA (Izmailov et al. 2018; Chen et al. 2021), AWP (Wu et al. 2020).
> * Generate additional data or create auxiliary examples. e.g. use DDPM data (Gowal et al., 2020; Rebuffi et al., 2021a), HAT (Rade et al., 2021)
> * Create effective data augmentation strategies to enhance robustness. e.g. CutMix (Yun et al. 2019; Rebuffi et al. 2021a), DAJAT (Addepalli et al. 2022)
> * Knowledge distillation. e.g. ARD (Goldblum et al., 2020b), RSLAD (Zi et al., 2021), IAD (Zhu et al., 2022)
>
> While WA and AWP concentrate on smoothing the model weight landscape, ADR focuses on smoothing the training labels. As their goals are distinct and not in conflict, direct comparisons between AT+ADR and AT+WA, or between AT+WA+ADR and AT+WA+AWP, may not be entirely fair. Their combined effects may offer a more comprehensive approach to improving adversarial training, addressing both the model's weight landscape and the label-smoothing aspects, bringing the overall robustness enhancement.
>
> **Q: The comparison with RobustBench in the paper is not fair, results on AT+ADR rather than AT+WA+AWP+ADR should be used. In this comparison, ADR achieves state-of-the-art robust performance on ResNet-18 but has a poor performance on WRN.**
>
> We do not agree that AT+ADR rather than AT+WA+AWP+ADR should be used for the RobsutBench comparison.   It is acknowledged that RobustBench, a benchmark for adversarial robustness, often includes methods that integrate several techniques rather than reporting solely on individual methods. As the common research directions for adversarial training that we provided in our previous reply, the methods from each direction can be combined as complementary to obtain better results.  Ultimately, the goal is to employ a combination that best suits the experimental objectives and the overall enhancement of adversarial robustness.
>
> For reference, Zhang et al. (2022), Rebuffi et al. (2021a), and Rade & Moosavi-Dezfooli (2022), report the results combined with WA.
> Addepalli et al. (2022), Chen & Lee (2021), and Jia et al. (2022)
> report their results combined with AWP.

---

> ### Author Response · Authors · 2023-11-21
> **Response for Reviewer VvcZ (continued)**
>
> **Q: The performance on WRN is not competitive enough.**
>
> We acknowledge that the performance reported on WRN may not be competitive enough in some cases. This phenomenon is primarily attributed to our deliberate decision to maintain consistency in experimental settings across different models, eliminating the need to report distinct hyperparameters for each model.
>
> To ensure uniformity, we employ the same $\tau$ and $\lambda$ schedule for both ResNet-18 and WRN-34-10, and the reported temperature and label interpolation factor are determined based on the evaluation of performance on ResNet-18. It is crucial to note that the chosen parameters may not represent the optimal fit for WRN-34-10, as the most effective parameters can vary based on architecture and specific settings, as previously mentioned in our limitations. We will investigate an automatic way to determine the optimal parameter in our future research.
>
> Due to we only have limited time and the substantial computational demands associated with adversarial training on expansive models like WRN, we only provide a full comparison on CIFAR-10 for reference. We can provide more experiment results in the final version of our paper.
> In the experiment, we keep $\lambda$  scheduling from 0.7 to 0.95 same as the original paper but adjust $\tau$ to anneal from 1.5 to 1 instead of the original 2.5 to 2.
>
> ***CIFAR-10***
> |   | AutoAttack  |   Standard Acc. |
> |---|---|---|
> | AT+ADR   (original) | 53.25 | 84.67 |
> | AT+ADR | 53.54 | 85.58 |
> | AT+WA+ADR   (original) | 54.10 | 82.93 |
> | AT+WA+ADR | 54.14 | 83.10 |
> | AT+WA+AWP+ADR   (original) | 55.26 | 86.11 |
> | AT+WA+AWP+ADR | 55.27 | 86.90 |
>
> An additional CIFAR-100 result when we anneal $\tau$ from 1.25 to 1 instead of 1.5 to 1.
>
> ***CIFAR-100***
> |   | AutoAttack  |   Standard Acc. |
> |---|---|---|
> | AT+ADR   (original) | 29.35 | 59.76 |
> | AT+ADR | 29.79 | 60.17 |
>
> A larger architecture generally requires a lower temperature to achieve its best performance, it may be because of their superior capability to learn the inter-class relationship within the data.
> We can observe that when selecting more suitable hyper-parameters for WRN-34-10, both robust and clean accuracy can be further enhanced.

---

> > ### Comment · Reviewer_VvcZ · 2023-11-22
> > **Response to authors**
> >
> > Dear authors
> > Thanks for your detailed explanation and clarification. The supplementary explanations resolve my confusions. Based on the experimental results provided, I reconsider the proposed method as valid and relatively innovative. Thus, I will raise the rating.

---

> > > ### Author Response · Authors · 2023-11-23
> > >
> > > We genuinely appreciate the reviewer's recommendation to accept our paper. Thank you very much for your time and efforts in reviewing.

---

### Official Review · Reviewer_YNYz · 2023-10-31

**Soundness:** 3 good
**Presentation:** 3 good
**Contribution:** 2 fair
**Rating:** 6
**Confidence:** 4

**Summary:**

This work proposes a novel adversarial training scheme where the ground-truth labels are rectified with an EMA teacher network. The experimental results show that the proposed method achieves a better accuracy-robustness trade-off with smaller overfitting gaps than the baselines. The proposed method can also be integrated with other AT approaches and brings further robustness boost.

**Strengths:**

- I appreciate the simplicity of the proposed method. It should be easy to implement and it can be used with other AT techniques.

- The paper presents a clear motivation for the proposed method with experimental results. I also find the idea behind inspiring as it is somewhat consistent with my understanding of AT, i.e., easier training (less adversarial/weaker signal) leads to better results.

- It is shown that the technique brings small additional training time.

- The paper is well-written with nice figures and a clear structure.

**Weaknesses:**

- Considering the simplicity of the method, I think the experiments are not very extensive. More datasets (e.g., ImageNet), attacks, and especially baseline AT methods should be considered.

- The self-distillation brings high memory cost, which I believe would be a main limitation for the practical use of the proposed method.

- It seems that this work is not the first to reveal that robust models are better calibrated, yet the authors conclude this finding as one of the major contributions of this paper.

**Questions:**

Did you use any pretrained models for initialization? I wonder how ADR works when using pretraining models.

---

> ### Author Response · Authors · 2023-11-20
> **Response for Reviewer YNYz**
>
> Dear reviewer YNYz,
>
> We thank you for your valuable comments. We are glad that you recognize our empirical contribution and find our paper well-written. We provide further clarifications on the concerns that you raise.
>
> **Q: Considering the simplicity of the method, I think the experiments are not very extensive. More datasets (e.g., ImageNet), attacks, and especially baseline AT methods should be considered.**
>
> Thank you for your suggestion. However, despite the apparent simplicity of ADR, it is essential to note that the training cost associated with ADR is comparable to other multi-step adversarial training methodologies, which are recognized for demanding computationally intensive processes. Our observations suggest that very few works employing multi-step adversarial training methodologies have presented results on the extensive ImageNet dataset. Instead of ImageNet, we present results on TinyImageNet-200, a representative subset, to underscore the practical effectiveness of our approach to real-world data.
> In our comparative analysis, the related works we compared (Rebuffi et al. 2021b; Yun et al. 2019; Rebuffi et al. 2021a; Addepalli et al. 2022; Li & Spratling 2023; Rade & Moosavi-Dezfooli, 2022; Gowal et al., 2020; Cui et al., 2021; Sehwag et al., 2022; Jia et al., 2022; Chen & Lee, 2021) refrain from experimenting with ImageNet and limit their baselines primarily to PGD-AT, TRADES with WA, and AWP. Consequently, we assert that the experiments we provide offer a comprehensive and representative baseline for evaluation.
>
> Acknowledging the constraints imposed by time and resource limitations during this rebuttal period, we regret our inability to present new results on additional datasets. However, we concur with the notion that augmenting experiments on larger datasets, coupled with diverse methods, would greatly benefit the research community. We commit to providing more comprehensive results in future work.
>
> **Q: The self-distillation brings high memory costs, which I believe would be a main limitation for the practical use of the proposed method.**
>
> While we acknowledge the additional memory costs associated with maintaining an EMA teacher for rectified distribution, it is crucial to underscore that the ADR method does not incur higher memory usage compared to previous techniques. Notably, ADR remains competitive in terms of performance without necessitating increased memory allocation. For instance, WA upholds a weight average copy of network parameters to attain a smoother model weight landscape for enhanced robustness. AWP also maintains an additional copy of the gradient direction at each step to calculate the updated direction for models. Consequently, ADR demonstrates competitive performance levels when deployed under equivalent resource constraints as its counterparts.
>
> **Q: It seems that this work is not the first to reveal that robust models are better calibrated, yet the authors conclude this finding as one of the major contributions of this paper.**
>
> In our analysis of Section 3, our findings indicate that robust models consistently demonstrate superior calibration and output stability across diverse input conditions. This aligns with the insights presented by Grabinski et al. (2022), as referenced in our paper. However, it is noteworthy that Grabinski et al. (2022) primarily conduct empirical analyses on a range of adversarially trained networks (comprising 71 models) without leveraging this characteristic to enhance adversarial robustness. In contrast, ADR is inspired by this observation and effectively enhances robustness performance.
>
> Notably, a comprehensive survey of existing literature reveals an absence of prior discussions on the phenomenon wherein adversarially trained networks consistently exhibit output consistency between adversarial images and their clean counterparts. Previous studies suggest the incorporation of dual targets during adversarial training, encouraging models to learn from both an adversarially trained model on adversarial images and a naturally trained model on normal images (Cui et al., 2021; Zhao et al., 2022). While seemingly intuitive in mitigating the trade-off between robustness and accuracy, our observations challenge the appropriateness of this approach. We contend that learning from two targets, one with high confidence from the normal model and the other with lower confidence from the adversarially trained model, is not conducive to achieving consistency for images with noise. Thus, we emphasize our contribution to these observations, underscoring the significance of designing a unified rectified label for all samples (both clean and adversarial images) used in adversarial training. We believe that this insight has the potential to benefit the research community dedicated to advancing adversarial robustness through the formulation of more effective training objectives.

---

> > ### Comment · Reviewer_YNYz · 2023-11-20
> >
> > > However, it is noteworthy that Grabinski et al. (2022) primarily conduct empirical analyses on a range of adversarially trained networks (comprising 71 models) without leveraging this characteristic to enhance adversarial robustness.
> >
> > I am not saying that this work is not different from previous works. I'm just saying that I think it's inappropriate to claim that "revealing robust models are better calibrated" is a contribution of this paper (maybe "confirming" is).
> >
> > > Notably, a comprehensive survey of existing literature reveals an absence of prior discussions on the phenomenon wherein adversarially trained networks consistently exhibit output consistency between adversarial images and their clean counterparts.
> >
> > Can you show the reference to this statement?
> >
> > > While seemingly intuitive in mitigating the trade-off between robustness and accuracy, our observations challenge the appropriateness of this approach...
> >
> > I understand that this is a valuable finding and an important motivation for the proposed method. But I think the current summarized contribution#1 omits the important part of this motivation. Therefore I would suggest refining this paragraph.

---

> > > ### Author Response · Authors · 2023-11-20
> > > **Reply for Reviewer YNYz**
> > >
> > > Thank you for your valuable suggestions,
> > >
> > > we refined our paper in the contribution paragraph with the following changes.
> > >
> > > * We adjust our contribution to confirm the calibration property of robust models instead of revealing it.
> > > * We strengthen our motivation with the finding that robust models exhibit output consistency for clean and adversarial examples.
> > >
> > > >Can you show the reference to this statement?
> > >
> > > We are not aware of studies explicitly emphasizing the aspect of output consistency in adversarially trained models as in our current work to the best of our knowledge. Therefore, we are unable to provide a reference for this specific aspect.

---

> > > > ### Comment · Reviewer_YNYz · 2023-11-21
> > > >
> > > > Thank you very much for your answers.
> > > >
> > > > > We are not aware of studies explicitly emphasizing the aspect of output consistency in adversarially trained models as in our current work to the best of our knowledge. Therefore, we are unable to provide a reference for this specific aspect.
> > > >
> > > > By "this statement" I meant "a comprehensive survey of existing literature reveals an absence of prior discussions on the phenomenon ..." So, I think at least you could show the title of this survey and preferably how it reveals the absence of the discussion.

---

> > > > > ### Author Response · Authors · 2023-11-21
> > > > > **Reply for Reviewer YNYz**
> > > > >
> > > > > We can provide some works that provide analyses of the model output distribution.
> > > > >
> > > > > * Confidence-Calibrated Adversarial Training: Generalizing to Unseen Attacks (Stutz et al. 2020) provides confidence histograms to compare their proposed CCAT model with AT. Their goal is to find a confidence threshold to distinguish the majority of adversarial examples.
> > > > > * Improving calibration through the relationship with adversarial robustness (Qin et al. 2021) made advancements in calibration by addressing adversarially unrobust inputs, focusing on predictive entropy when models encounter previously unseen data.
> > > > >
> > > > > While these works engage in some output distribution analyses,  they did not discuss the consistency of output distribution for the adversarially trained models.

---

> > > > > > ### Comment · Reviewer_YNYz · 2023-11-23
> > > > > >
> > > > > > Now I understand that "a comprehensive survey" refers to a survey done by the authors. I was thinking that there's a survey paper.
> > > > > >
> > > > > > After I read other reviews and the authors' responses, I decided to keep my score as borderline accept.

---

> > > > > > > ### Author Response · Authors · 2023-11-23
> > > > > > >
> > > > > > > We greatly appreciate the reviewer's ongoing engagement with our paper and the feedback you have provided.
> > > > > > > Thank you for your time and efforts in reviewing our work.

---

> ### Author Response · Authors · 2023-11-20
> **Response for Reviewer YNYz (continued)**
>
> **Q: Did you use any pre-trained models for initialization? I wonder how ADR works when using pretraining models.**
>
> Thanks for your interesting suggestion, we leverage pre-trained models for the initialization of the ADR process. Specifically, we employ ResNet-18 on the CIFAR-10 dataset, adhering to all other parameter settings outlined in Section 5.1. The key variation lies in the initialization of the model with either a standard or PGD-AT pre-trained model, followed by subsequent ADR training. Our reporting is based on the optimal checkpoint obtained during our comprehensive evaluation.
> Moreover, to offer a more comprehensive view of the observed trends, we present line charts depicting the performance trajectories with both clean and robust accuracy in Figure 10. These visual representations are included in the Appendix L of our updated paper for enhanced clarity and accessibility.
>
> The experimental findings reveal a consistent trend wherein ADR consistently outperforms the baseline in terms of robust accuracy, regardless of whether it is initialized with a pre-trained model or not. The utilization of either a standard model or a PGD-AT pre-trained weight as an initialization fails to further augment the maximum robustness achievable through ADR. Nevertheless, it is noteworthy that employing an AT pre-trained checkpoint results in a notable enhancement of standard accuracy by 1.25% when compared to training with random initialization. This outcome underscores the potential for mitigating the accuracy-robustness tradeoff, indicating the feasibility of achieving improved performance by utilizing an AT pre-trained model during the initialization phase.
>
> |   | AutoAttack  | Standard Acc. |
> |---|---|---|
> | AT|48.81 |82.52 |
> |AT + ADR|50.38|82.41|
> |AT + ADR + standard pre-trained initialization | 49.45 | 82.22 |
> |AT + ADR + AT pre-trained initialization | 50.11 | 83.66 |

---

> > ### Comment · Reviewer_YNYz · 2023-11-20
> >
> > Thank you very much for your response.
> >
> > > Nevertheless, it is noteworthy that employing an AT pre-trained checkpoint results in a notable enhancement of standard accuracy by 1.25% when compared to training with random initialization
> >
> > I want to make sure if this means AT+AT pre-trained initialization results in 82.52%+1.25%=83.77% standard accuracy. If so, what is the AA accuracy?
> >
> > I also would like to know how you obtain the AT pre-trained initialization.

---

> > > ### Author Response · Authors · 2023-11-20
> > > **Reply for Reviewer YNYz**
> > >
> > > Thanks for asking this question.
> > >
> > > Let us clarify that we compare the standard accuracy of AT+ ADR + AT pre-trained initialization with AT + ADR as it seems to be more fair to use it as a control group when only one factor is modified. 82.41% + 1.25% = 83.66%
> > > The robust accuracy is 50.11%, same as what we reported in the table.
> > >
> > > We obtained the AT pre-trained checkpoint by simply training by ourselves. To be specific, we use the checkpoint obtained in AT (first row of the table) as initialization and further go through the training with ADR.

---

### Official Review · Reviewer_EopP · 2023-10-31

**Soundness:** 2 fair
**Presentation:** 2 fair
**Contribution:** 3 good
**Rating:** 6
**Confidence:** 3

**Summary:**

The paper investigates the phenomenon of robust overfitting and reports that robust models exhibit outputs more calibrated compared to standard models. It further proposes a label smoothing scheme for mitigating overfitting in adversarial training via employing model weight averaging, annealed interpolation and softmax temperature. Experimental results indicate robustness gains and a reduction of the severity of robust overfitting.

**Strengths:**

- The experiments performed are extensive and incorporates several baseline methods for comparison.
- Modifying self-distillation EMA (weight averaging) with annealed interpolation and softmax temperature is an interesting idea.

**Weaknesses:**

- While the method is shown to increase the robustness and mitigates overfitting, other experimental results seem to lack significance for drawing conclusions (weight loss lanscape, effectiveness of temperature and interpolation parameters).
- The relation to other similar methods investigated is not clear (see questions).

**Questions:**

- In Table 2, test accuracies of ADR combined with WA and AWP are presented. From the results it seems that each method contributes a (roughly) similar amount of robustness gain. Could the authors comment on whether they consider ADR, WA, AWP to be complementary?
- In the motivation (subsection 4.1.) it is stated that the aim is 'to design a label-softening mechanism that properly reflects the true distribution'. In the presented approach the teacher network provides the probability distribution with which the one-hot label is interpolated. Can the authors comment on why the teacher network (model weight average of student) reflects the 'true distribution'?
- The results presented in Figure 5 show the results of a hyperparameter study for the interpolation and temperature parameter. It is concluded that annealing both parameters is beneficial for robustness. Do the authors consider the differences (e.g. in the row on $\lambda$ annealing) significant enough to draw such conclusions?

---

> ### Author Response · Authors · 2023-11-20
> **Response for Reviewer EopP**
>
> Dear reviewer EopP:
>
> Thank you for your reviews, we are glad that you find our idea interesting. We address your questions in the following.
>
> **Q: While the method is shown to increase the robustness and mitigate overfitting, other experimental results seem to lack significance for drawing conclusions (weight loss landscape).**
>
> Several prior studies, including works by Wu et al. (2020) and Stutz et al. (2021), have established a correlation between the flatness of a model's weight landscape and its adversarial robustness. A smooth weight landscape indicates that the model produces a more stable output when subjected to slight weight perturbations, resulting in minimal changes in loss, as illustrated in Figure 4. Diverse methodologies have been proposed in the literature to enhance robustness by inducing a flatter weight landscape. (Wu et al. 2020; Chen et al. 2021; Gowal et al. 2020; Zhang et al. 2019). It is noteworthy that our baseline methods such as WA and AWP inherently contribute to landscape smoothness through their respective designs. In this context, the observation that ADR achieves a flatter weight loss landscape provides compelling evidence supporting our claim of achieving enhanced robustness. This alignment with the established relationship between weight landscape flatness and robustness further reinforces the efficacy of ADR in promoting a more resilient model.
>
> **Q: The results presented in Figure 5 show the results of a hyperparameter study for the interpolation and temperature parameters. It is concluded that annealing both parameters is beneficial for robustness. Do the authors consider the differences (e.g. in the row on annealing) significant enough to draw such conclusions?**
>
> While we acknowledge that the variations in the number of occurrences in each row and column in Figure 5 may not be pronounced, a discernible trend emerges when implementing annealing in $\tau$. Notably, the clean and robust accuracy consistently outperforms scenarios where the temperature is fixed at a specific value across most cases as $\lambda$ varies. Similarly, the highest robust accuracy is attained when annealing in $\lambda$ instead of maintaining a fixed $\lambda$ value while varying the temperature. It is noteworthy that, although there may be a slight sacrifice in clean accuracy when annealing in $\lambda$ during training, such trade-offs are inherent, given the well-established inverse relationship between model accuracy and robustness. Thus, we conclude that annealing in both temperature and label interpolation factors proves advantageous in achieving superior overall performance.
>
> **Q: Could the authors comment on whether they consider ADR, WA, and AWP to be complementary?**
>
> In the realm of adversarial defense, we can encapsulate the directions of research aimed at enhancing the efficacy of adversarial training into the following key areas:
> * Different training objectives. e.g. PGD-AT (Madry et al., 2018), TRADES (Zhang et al., 2019).
> * Encourage a flatter model-weight landscape. e.g.  WA (Izmailov et al. 2018; Chen et al. 2021), AWP (Wu et al. 2020).
> * Generate additional data or create auxiliary examples. e.g. use DDPM data (Gowal et al., 2020; Rebuffi et al., 2021a), HAT (Rade et al., 2021)
> * Create effective data augmentation strategies to enhance robustness. e.g. CutMix (Yun et al. 2019; Rebuffi et al. 2021a), DAJAT (Addepalli et al. 2022)
> * Knowledge distillation. e.g. ARD (Goldblum et al., 2020b), RSLAD (Zi et al., 2021), IAD (Zhu et al., 2022)
>
> We provided an introduction to those methods in both Section 2 and Appendix A.
>
> Given the straightforward design of ADR, it becomes feasible to integrate ADR with other adversarial training methodologies as complementary approaches. The results of these combinations are presented in Table 1 for ADR with different training objectives (PGD-AT, TRADES), in Table 2 for ADR with flatter weight landscape methods (WA, AWP), and in Table 3 for ADR in conjunction with generated data (DDPM). The integration of ADR with these complementary methods demonstrates discernible gains in robustness.
>
> While it is acknowledged that data augmentation-based methods often necessitate intricate hyperparameter tuning and extended training schedules due to their reliance on diverse augmentation techniques, such complexities are not the focus of our work. Consequently, ADR is not combined with these methods in our experiments. Additionally, knowledge distillation-based methods, which modify training labels through the knowledge of pre-trained networks, are not suitable for combination with ADR. Instead, we provide explicit comparisons in Table 3 and Appendix Table 8, establishing ADR as a more effective and resource-efficient methodology.

---

> ### Author Response · Authors · 2023-11-20
> **Response for Reviewer EopP (continued)**
>
> **Q: In the motivation (subsection 4.1.) it is stated that the aim is 'to design a label-softening mechanism that properly reflects the true distribution'. In the presented approach the teacher network provides the probability distribution with which the one-hot label is interpolated. Can the authors comment on why the teacher network (model weight average of students) reflects the 'true distribution'?**
>
> Thanks for asking this question. Let us clarify that our primary objective is to design a label-softening mechanism that accurately reflects the true underlying distribution for each example. It's important to note that the teacher EMA network does not precisely mirror this ideal distribution from the outset. Instead, it progressively refines its approximation towards the goal of achieving both increased robustness and accuracy as the training process advances. Acknowledging the potential for confusion in our original language, we have revised the corresponding paragraph in our updated paper to ensure a more accurate and clear representation of our intentions.

---

> > ### Comment · Reviewer_EopP · 2023-11-22
> >
> > Many thanks to the authors for addressing my questions. The clarifications helped my understanding and I encourage the authors to integrate them into the paper. I updated my score.

---

> > > ### Author Response · Authors · 2023-11-23
> > >
> > > We thank the reviewer for the endorsement! Your comments and suggestions are instrumental to our work.

---

### Official Review · Reviewer_NJvp · 2023-11-01

**Soundness:** 3 good
**Presentation:** 3 good
**Contribution:** 3 good
**Rating:** 6
**Confidence:** 4

**Summary:**

This paper proposes Annealing Self-Distillation Rectification (ADR), an improved adversarial training (AT) method that emphasizes the rectification of the labels used in AT. It is found that the outputs of robust and non-robust models are distributionally different in several aspects, and it is argued that labels rectified in a noise-aware manner can better reflect the output distribution of a robust model. Hence, the proposed ADR uses the interpolation between the one-hot labels and the outputs of an EMA teacher to produce the rectified distributions, which replace the one-hot labels used in existing AT methods. Experimental results suggest that ADR can achieve state-of-the-art robust accuracy.

**Strengths:**

1. The proposed method ADR is intuitive and can be easily integrated into different AT methods.

2. Experimental results suggest that ADR can achieve significant improvement over the baseline and superior robust accuracy to existing methods.

3. The details of the method and the experiments are clearly stated, and the source code is also provided.

**Weaknesses:**

1. While Section 3.2 provides some insightful observations, it is not very clear how they are reflected by the design of the proposed ADR. Particularly, Section 4.1 mentions that robust models should "generate nearly random probability on OOD data" and "demonstrate high
uncertainty when it is likely to make a mistake". However, there is no direct evidence (analytical or empirical) for how ADR may help achieve these properties. Instead, it seems that the motivation for ADR mostly comes from the previous works (like those cited in Section 4.1) that suggested the importance of label rectification in AT.

2. In Table 3, it is shown that using DDPM data can improve the robust accuracy of WRN-34-10 trained via AT+ADR, but at the cost of a significant decrease in standard accuracy. This may be undesired since augmenting the training set with DDPM data can improve both clean and robust accuracy for AT according to (Rebuffi et al., 2021a). There should be some explanations or discussions on this issue.

3. The texts in Figure 2 may be too small, which can be difficult to read when printed out.

**Questions:**

1. Are the robust models trained via ADR more conformed to the empirical findings in Section 3.2, as compared with vanilla AT?

2. Should the rectified labels be assigned to adversarial images only, or both clean and adversarial images? Considering that different AT methods use different targets for clean and adversarial images (e.g., PGD-AT and TRADES), this can be an important question when one would like to apply ADR to other AT methods.

---

> ### Author Response · Authors · 2023-11-20
> **Response for Reviewer NJvp**
>
> Dear reviewer NJvp:
>
> Thank you for your valuable comments. We are glad you find our method intuitive and recognize our achievement in obtaining effective robustness. We clarify your concerns below:
>
> **Q: While Section 3.2 provides some insightful observations, it is not very clear how they are reflected by the design of the proposed ADR. Particularly, Section 4.1 mentions that robust models should "generate nearly random probability on OOD data" and "demonstrate high uncertainty when it is likely to make a mistake". However, there is no direct evidence (analytical or empirical) for how ADR may help achieve these properties. Instead, it seems that the motivation for ADR mostly comes from the previous works (like those cited in Section 4.1) that suggested the importance of label rectification in AT.**
>
> In our observations in Section 3.2, we discover that adversarially trained networks consistently exhibit output consistency between adversarial images and their clean counterparts. Some previous studies suggest the incorporation of dual targets during adversarial training, encouraging models to learn from both an adversarially trained model on adversarial images and a naturally trained model on normal images (Cui et al., 2021; Zhao et al., 2022). While seemingly intuitive in mitigating the trade-off between robustness and accuracy, our observations challenge the appropriateness of this approach. We contend that learning from two targets, one with high confidence from the normal model and the other with lower confidence from the adversarially trained model, is not conducive to achieving consistency for images with noise. Therefore, we propose composing a rectified label in ADR and urge that all images within the $l_p$ norm neighborhood of a given input should be targeted using the rectified label rather than the one-hot one. The observations in Section 3 motivate us to design ADR that effectively prevents the network from becoming over-confident and provides semantically aware smoothing rather than a uniform one.
>
> **Q: Are the robust models trained via ADR more conformed to the empirical findings in Section 3.2, as compared with vanilla AT?**
>
> Thanks for asking, we provide our entropy distribution analysis on the ADR-trained model in the updated paper of Appendix K.
>
> From Figure 9a, we can see that the ADR model trained on CIFAR-10 is more uncertain than the model trained with PGD-AT when encountering out-of-distribution data (CIFAR-100). It supports our claim that the robust model should not be over-confident when seeing something it has never seen before.
> From Figure 9b and Figure 9c, we can observe that the ADR-trained model exhibits similar output distribution as the model trained with PGD-AT (Figure 2b and Figure 2d), except that the ADR-trained model exhibits higher entropy levels in general. The reason behind this phenomenon is that ADR builds a sofer target as training labels, so the output of the ADR model will also be smoother, resulting in higher entropy levels on average.
>
> **Q: In Table 3, it is shown that using DDPM data can improve the robust accuracy of WRN-34-10 trained via AT+ADR, but at the cost of a significant decrease in standard accuracy. This may be undesired since augmenting the training set with DDPM data can improve both clean and robust accuracy for AT according to (Rebuffi et al., 2021a). There should be some explanations or discussions on this issue.**
>
> We acknowledge that the observed drop in standard accuracy for WRN-34-10 when trained with additional DDPM data is not desirable. This phenomenon is primarily attributed to our deliberate decision to maintain consistency in experimental settings across different models, eliminating the need to report distinct hyperparameters for each model.
>
> To ensure uniformity, we employ the same $\tau$ and $\lambda$ schedule for both ResNet-18 and WRN-34-10, and the reported temperature and label interpolation factor are determined based on the evaluation of performance on ResNet-18. It is crucial to note that the chosen parameters may not represent the optimal fit for WRN-34-10, as the most effective parameters can vary based on architecture and specific settings, as previously mentioned in our limitations. Consequently, the undesired drop in clean accuracy for WRN-34-10 in Table 3 can be attributed to the suboptimal parameter selection for this particular architecture. We will investigate an automatic way to determine the optimal parameter in our future research.

---

> ### Author Response · Authors · 2023-11-20
> **Response for Reviewer NJvp (continued)**
>
> **Q: The texts in Figure 2 may be too small, which can be difficult to read when printed out.**
>
> Thank you for the suggestion, we provided adjusted figures that have larger texts in our revised paper.
>
> **Q: Should the rectified labels be assigned to adversarial images only, or both clean and adversarial images? Considering that different AT methods use different targets for clean and adversarial images (e.g., PGD-AT and TRADES), this can be an important question when one would like to apply ADR to other AT methods.**
>
> The rectified labels should be assigned to both clean and adversarial images. Our observations, detailed in Section 3.2, lead us to summarize that a robust model should consistently produce outputs that align for both clean and adversarial data. In essence, within the $l_p$ norm neighborhood of a given input, all images should be associated with a singular smooth and rectified label. Consequently, we propose the replacement of both hard labels used in PGD-AT for adversarial images and TRADES for clean images with the rectified labels generated by the ADR process. This adjustment ensures a unified and consistent labeling approach across both clean and adversarial instances, aligning with our overarching objective of enhancing model robustness.

---

> > ### Comment · Reviewer_NJvp · 2023-11-22
> >
> > Thanks for your valuable responses. Most of my concerns have been addressed, and I appreciate the improved clarification of the motivation and contributions in the responses to all reviewers. As for the suboptimal clean accuracy in some experiments, there is still a lack of convincing evidence that this is not a weakness of the proposed method, although I understand that diving into the tuning of the hyperparameters for WRN-34-10 can potentially tackle this issue, and this is not likely to be finished during this discussion period, especially the experiment using DDPM data. Therefore, I will consider raising my final score and I hope that better results can be presented in the final version of the paper.

---

> > > ### Author Response · Authors · 2023-11-23
> > >
> > > Thank you for your constructive feedback.
> > >
> > > We are glad that most of your concerns are addressed.
> > > Regarding the suboptimal clean accuracy in some experiments, we appreciate your understanding of the difficulty of tuning hyperparameters on WRN models with additional DDPM data within the discussion period.
> > > However, we have a preliminary result to provide as the experiment has just been completed. In this setting, we set $\tau$ anneal from 3 to 1.5 instead of the original 1.5 to 1.
> > >
> > > ***CIFAR-100 + DDPM***
> > >
> > > | | AutoAttack | Clean Acc. |
> > > |--|--|--|
> > > |AT+ADR (original) | 32.19 | 59.60 |
> > > |AT+ADR | 32.60 | 60.48 |
> > >
> > > From the table, we can observe that both robust and clean accuracy can be further enhanced by selecting more suitable hyper-parameters. Still, the result is just preliminary and there is still room for clean accuracy improvement. We would like to assure you that we are dedicated to providing more comprehensive and conclusive results.
> > >
> > > We sincerely appreciate your consideration in reevaluating the final score, and we are committed to delivering a high-quality and thoroughly validated final version of the paper.
> > >
> > > Thank you again for your valuable feedback.

---

### Meta-Review · Area_Chair_T6qV · 2023-12-07

**Metareview:**

Summary: The submission proposes the generation of soft labels instead of hard one-hot labels to help obtain more robust models. The method is integrated with known adversarial training techniques to help avoid misclassification in the presence of adversarial noise.

+ The paper is clearly written.
+ The proposed method is simple, so it could be easily incorporated by others.
+ The experiments, although limited, suggest the method is effective.

- There is limited novelty.
- The experimental setup is not extensive. Detailed reviews provide suggestions on improving this.

**Justification For Why Not Higher Score:**

- There is limited novelty.
- The experimental setup is not extensive. Detailed reviews provide suggestions on improving this.

**Justification For Why Not Lower Score:**

+ The paper is clearly written.
+ The proposed method is simple, so it could be easily incorporated by others.
+ The experiments, although limited, suggest the method is effective.

---

### Decision · Program_Chairs · 2024-01-16

Accept (poster)